# A substitutional quantum defect in WS$_2$ discovered by high-throughput computational screening and fabricated by site-selective STM manipulation

John C. Thomas [1,2,3,12] ✉, Wei Chen [4,12], Yihuang Xiong[3,12], Bradford A. Barker[5], Junze Zhou[1], Weiru Chen [3], Antonio Rossi [1,2,6], Nolan Kelly[5], Zhuohang Yu [7,8], Da Zhou [9], Shalini Kumari [7,8], Edward S. Barnard [1], Joshua A. Robinson[7,8,9,10], Mauricio Terrones[7,8,9,10], Adam Schwartzberg [1], D. Frank Ogletree [1], Eli Rotenberg [6], Marcus M. Noack [11], Sinéad Griffin [1,2], Archana Raja [1,2], David A. Strubbe [5], Gian-Marco Rignanese [4], Alexander Weber-Bargioni[1,2] ✉ & Geoffroy Hautier [3] ✉

Point defects in two-dimensional materials are of key interest for quantum information science. However, the parameter space of possible defects is immense, making the identification of high-performance quantum defects very challenging. Here, we perform high-throughput (HT) first-principles computational screening to search for promising quantum defects within WS$_2$, which present localized levels in the band gap that can lead to bright optical transitions in the visible or telecom regime. Our computed database spans more than 700 charged defects formed through substitution on the tungsten or sulfur site. We found that sulfur substitutions enable the most promising quantum defects. We computationally identify the neutral cobalt substitution to sulfur (Co$_S^0$) and fabricate it with scanning tunneling microscopy (STM). The Co$_S^0$ electronic structure measured by STM agrees with first principles and showcases an attractive quantum defect. Our work shows how HT computational screening and nanoscale synthesis routes can be combined to design promising quantum defects.

Point defects in semiconductors are considered as building blocks for quantum information science (QIS) applications. Optically-active quantum defects (OQDs) can be used in quantum sensing, memory, and networks[1–4]. The performance of an OQD depends on its fundamental properties and limitations that can vary across defects[5,6]. Certain defects, such as the silicon-divacancy center in diamond, show robust optical coherence but low spin coherence, while the NV$^-$ center in diamond shows high spin coherence but lower optical coherence[7,8].

The identification of OQDs in a specific host with optimal spin, optical, and electronic properties is essential to the development of QIS applications.

Two-dimensional (2D) materials, particularly transition metal dichalcogenides (TMDs), provide an enormous phase space of functionality with tunable and exceptional spin, optical and electronic properties[9–17]. Additionally, as materials are reduced from bulk to lower dimensionality, the spin-coherence lifetime of an OQD is

expected to increase[18]. $WS_2$, specifically, is a highly modifiable TMD that has been predicted to have long spin coherence times ($T_2$ of ~11 ms)[18,19]. A decisive factor for an OQD is the appearance of in-gap localized states making it important to understand and measure the electronic levels induced by a defect in a given 2D host. While a number of techniques can routinely resolve the atomic lattice, the electronic levels introduced by the defect in the host are not easily accessible by most experimental techniques. However, scanning tunneling microscopy (STM) and scanning tunneling spectroscopy (STS) can probe atomic-scale defects at the required length scale[20,21]. This has been used to characterize many defects in 2D materials, e.g., carbon radical dopants, chalcogen vacancies, oxygen substitutions, and a variety of metal substitutions[15,16,20,22–24]. Next to these experimental developments, first-principles approaches have been successfully used to compute and understand the properties of quantum defects in bulk semiconductors and 2D hosts[25–28]. First principles techniques have even been used to suggest OQDs in 2D materials, but these studies have remained targeted on a few defects and have not browsed the large elemental space of possible defects[29–33].

Here, we use first principles high-throughput (HT) computing to build a database of point defects in $WS_2$ considering all possible substitutional defects from 57 elements, aiming to accelerate the exploration of defect chemical space in $WS_2$[34]. We use this database to identify a handful of promising defects and show that the substitution of cobalt for sulfur ($Co_S$) in $WS_2$ is especially appealing. First principles computations indicate that the neutral $Co_S$ shows several localized levels in the band gap, spin multiplicity, and potential for bright

telecom emission. This defect is then synthesized in situ, and examined with STM/STS, and the measured energy levels confirm and benchmark the theoretical predictions, which highlights an attainable two-level quantum system.

## Results

### High-throughput search

A greatly sought-after electronic structure for an OQD involves two localized defect levels (one occupied, the other unoccupied) well within the bandgap[35]. This requires a precise matching of defect and band edge levels. Additionally, the optical transition between these defect levels should be bright and exhibit large transition dipole moments (TDMs). While having localized defect levels within the band gap is not in itself necessary for developing OQDs, this electronic structure has advantages in terms of brightness and robustness versus temperature[36]. With the 2.4 eV electronic band gap for $WS_2$, finding defect levels that are at the same time isolated within the band gap and with transitions in the telecom or visible range (from 750 meV up to 2 eV) should be achievable. However, identifying defects that could act as an OQD within $WS_2$ is challenging.

To search for such a defect, we have built a database with the computed electronic structure of 757 charged point defects in $WS_2$ considering either the tungsten ($M_W$) or sulfur ($M_S$) substitution site (see Fig. 1a). All the elements from the periodic table are used with the exception of rare-earths and transuranides. We start our screening by computing the relaxed structure and formation energies of the defect in multiple charge states within Density Functional Theory (DFT) in the

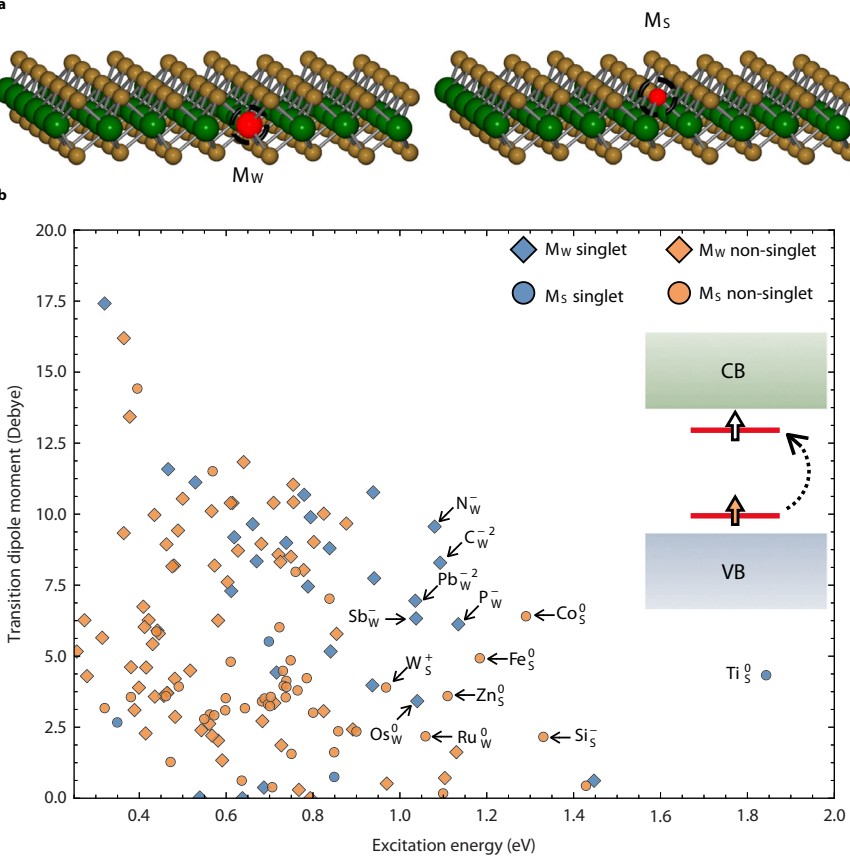

**Fig. 1 | Two-level quantum defect screening in $WS_2$. a** Two defect configurations that are considered in this work: substitution on the W site ($M_W$, red) is shown on the left and on S site ($M_S$, red) is depicted on the right. W atoms are colored green and S atoms are displayed in yellow. **b** Transition dipole moment vs. single-particle excitation energy at the single-shot PBE0. The marker and color scheme stand for the defect structure and whether the ground state is singlet or not. Each point stands for a charge defect that is thermodynamically stable within a certain Fermi level ($E_F$) range in the band gap, and with electronic structures that possess two localized defect levels within the band gap, as shown in the inset. Below the conduction band (CB, light green) and above the valence band (VB, light blue), a filled state (orange arrow) to empty state (white arrow) transition is shown.

generalized gradient approximation (GGA). Single-particle energies and band gaps are notoriously underestimated within DFT and one of the gold standards in defect computation is to use hybrid functionals such as PBE0 which adds a fraction of Fock exchange to the GGA functional[26,37].

Recently, we have shown that for 2D materials, using a modified fraction of Fock exchange for the defect and the host is not adequate and we use here an approach combining a different amount of Fock exchange for defect levels and band edges (see Methods)[37]. The use of hybrid functionals leads to a significantly higher computational cost and can preclude broad screening. Here, we accelerate the hybrid computation by fixing the wave function from DFT and applying the hybrid functional Hamiltonian from PBE0[38]. This single-shot PBE0 approach (or PBE0$_0$) is similar to the single-shot $GW$ ($G_0W_0$) approach and enables single-particle energy predictions that are much improved compared to DFT at a minimal computational overhead, which we have used for defects in silicon[38].

Our computational database includes formation energies, spin state, and single-particle electronic energy levels for all the possible charged defects. It also contains the TDMs between these single-particle levels indicating optical transition brightness. We use this database to search for attractive OQD candidates. Point defects in semiconductors can have different charge states depending on the Fermi level ($E_F$). Certain charge states are not stable for any $E_F$ within the band gap. While we do not study how a given $E_F$ can be achieved (e.g., through doping or gating) we only consider defects in a charge state that is stable for a range of $E_F$ within the band gap. In addition, we focus on charged defects with possible optical transitions between defect levels localized within the band gap. No criteria on the formation energy other than the need for the charged defect to have a range of $E_F$ in which it is stable was applied in our screening. For these defects, we evaluate their single-particle excitation energies and TDMs. Figure 1b shows the TDM versus excitation energy for all defects. We differentiate between $M_S$ and $M_W$ defects as well as singlets and multiplets. Few defects show high brightness (with a TDM of at least 2.5 D) and an excitation energy within the telecom or visible range (single-particle excitation energy > 750 meV) (see Supplementary Table 1 and Supplementary Figs. 1 and 2 for a full list with their single-particle levels). We identify a series of potential singlet OQDs that could act as single-photon emitters and are formed through the substitution of W with a main group element: $Sb_W^{-1}$, $P_W^{-1}$, $Pb_W^{-2}$, $N_W^{-1}$, and $C_W^{-2}$. Only two transition metal defects appear as promising singlets: $Os_W$ and $Ti_S$. For quantum applications, defects that possess a nonzero spin are often desirable and are called spin-photon interfaces[4,39–41]. In WS$_2$, spin multiplet defects only appear through sulfur substitution: $Co_S^0$, $Fe_S^0$, $Zn_S^0$, $Si_S^{-1}$, and $W_S^{+1}$ except for $Ru_W^0$. The $W_S$ defect has been suggested

as an OQD by Tsai et al. as well, but in the zero charge state[29]. Notably, common substitutional defects to tungsten in WS$_2$: Re, V, Nb, Mo, and Cr, do not show an adequate electronic structure (see Supplementary Fig. 3)[20,42–45]. They all have at most one level in the band gap of the substitutional $d$ orbital character that is slightly above the valence band edge (V) or below the conduction band (Cr and Re). They are only excitable optically through a transition between a localized defect state and a delocalized band level forming a bound exciton[36]. Our findings agree with experimental results from photoluminescence or STS on $M_W$ defects[20,42,44,45].

While substitutional transition metals on W sites are easy to synthesize[11,46], our screening results show that this is not the most promising approach for OQD discovery. All our candidate transition metal OQDs except $Ru_W^0$ and $Os_W^0$ show up instead as $M_S$. Figure 2a shows the different electronic structures for $M_W$ and $M_S$ in a molecular orbital diagram picture when M is a transition metal[47]. For both substitutions, the $d$ orbitals of the defect mix with either sulfur ($M_W$) or tungsten ($M_S$) forming bonding and anti-bonding states separated by $\Delta_{AB}$. Additionally the different $d$ orbitals are split into three groups: $(d_{xz}, d_{yz})$, $(d_{xy}, d_{y^2-x^2})$ and $d_{z^2}$ with an energy $\Delta d$ according to crystal field splitting theory. For the sake of simplicity, we assume here a C$_{3v}$ and D$_{3h}$ point group respectively for the $M_S$ and $M_W$ defects, where lower symmetry through Jahn-Teller distortions are also possible. We performed bonding analysis, and determined density of states, for all 3$d$ transition metal defects (Supplementary Fig. 4) and we observed a smaller splitting between bonding and anti-bonding states for $M_S$ versus $M_W$ ($\Delta_{AB}$). This can be rationalized by the different atomic positions of sulfur and tungsten orbitals. Additionally, the splitting between $d$ orbitals ($\Delta d$) is higher for $M_S$ versus $M_W$. Figure 2b shows the positions of bonding and anti-bonding molecular orbitals across the 3$d$ series ($M_W$ (blue) and $M_S$ (yellow) in the neutral charge state), where 3$d$ atomic orbitals shift to lower energy from Ti to Cu. $M_S$ substitutions show a clear advantage in terms of a smaller $\Delta_{AB}$ and larger $\Delta_d$, which leads to $d$–$d$ transitions in the telecom or visible range and enables more potential for OQDs with two levels localized in the band gap.

## Candidates

While our analysis shows that within gap $d$–$d$ transitions are more likely in $M_S$ and rationalizes why there are still differences between $M_S$ defects. Figure 1 shows that $Co_S^0$ is by far the most attractive OQD considering its non-singlet (doublet) spin multiplicity, its large excitation energy, and its transition dipole moment. We compute the electronic structure and formation energy for this $Co_S^0$ defect within full-fledged PBE0 computations including structural relaxation and self-consistency. We plot the defect formation energy for different charge states of Co$_S$ versus $E_F$ in Fig. 3a. The defect is stable in its zero

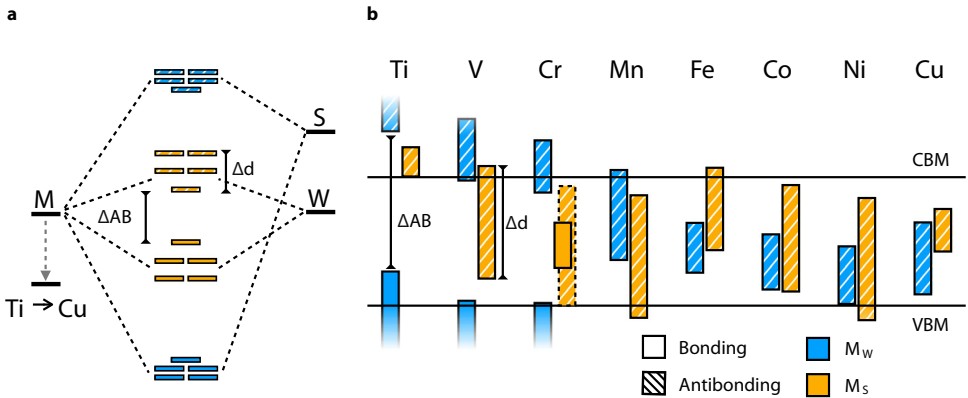

**Fig. 2 | Molecular orbital trend within the 3$d$ transition metal series for $M_S$ and $M_W$ defects. a** The molecular orbital diagram shows the splitting between anti-bonding and bonding state ($\Delta_{AB}$) as well as the splitting with $d$ orbitals ($\Delta d$) for a typical $M_W$ and $M_S$ defect. **b** A schematic of the bonding and anti-bonding state for different 3$d$ transition metals in $M_W$ (blue) and $M_S$ (yellow) positions. The conduction band minimum (CBM) and valence band maximum (VBM) are drawn as black lines.

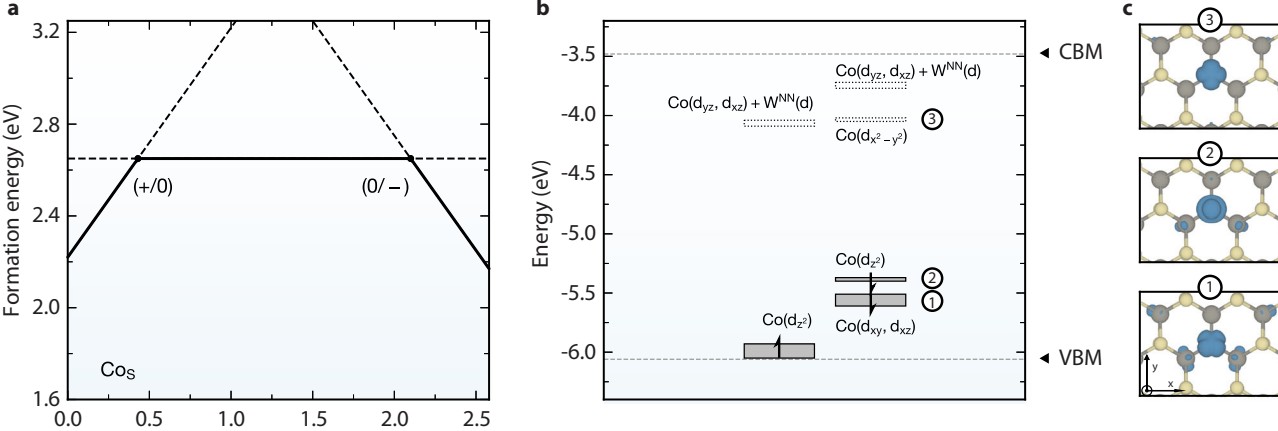

**Fig. 3 | Thermodynamic charge transition levels and electronic structure of Co$_S$. a** Formation energy of Co$_S$ as a function of Fermi level for the neutral and the two charged states. The charge transition levels, i.e., (+/0) and (0/−), are referenced to the band-edge positions of pristine WS$_2$ as obtained with PBE0 incorporating 22% of Fock exchange PBE0(0.22). **b** Orbital diagram of the localized defect states for neutral Co$_S$. Resonant states within the valence band and conduction band manifolds are not depicted. The characters of the localized states are indicated by the specific $d$ orbitals of the Co atom [e.g., Co($d_{z^2}$) as the highest occupied state in the minority channel] and, if any, the $d$ orbitals of the W atoms in the nearest neighbor (W$^{NN}$). The occupied (unoccupied) states are shown by the filled (empty) rectangles, the height of which indicates the degree of dispersion. The localized electrons in the majority (minority) channel are indicated by the arrows pointing up (down). The band-edge positions (in horizontal dashed lines) refer to those of the pristine WS$_2$ obtained with PBE0 (0.22). Energies are referenced to the vacuum level. SOC is not taken into account for the localized defect states. **c** Top view of the charge density (in blue) for the three Co$_S^0$ defect states as indicated in (**b**). The W and S atoms are represented by the gray and yellow spheres, respectively. The isovalue is 0.001 e/Å$^3$.

charge state spanning a large $E_F$ range. The two thermodynamic charge transition levels correspond to (+/0) at 0.4 eV above the valence band maximum (VBM) and (0/−) at 0.4 eV below the conduction band minimum (CBM).

The electronic structure of the neutral Co$_S^0$ is shown in Fig. 3b. A full description of the electronic structure for all three charge states is given in Supplementary Fig. 5. The neutral defect undergoes a Jahn-Teller distortion towards the C$_s$ symmetry. While there is significant mixing with the host, the projection on the Co-3$d$ orbitals is provided in Fig. 3b with the wavefunctions illustrated in Fig. 3c. The defect shows occupied $d_{xy} + d_{xz}$ and $d_{z^2}$ states well within the band gap that can be excited to the unoccupied $d_{x^2-y^2}$ state below the conduction band. The lowest energy transition is between the $d_{z^2}$ and $d_{x^2-y^2}$ states and sits at a 1.4 eV difference in single-particle energies and shows a TDM of 1.2 D. All these values are obtained from full PBE0 but confirm the prediction from our screening at the single-shot level. The zero-phonon lines (ZPL) associated with this transition are computed within the constrained-occupation DFT by imposing the electron occupation (or needed unoccupation) of the $d_{z^2}$ and $d_{x^2-y^2}$ states and relaxing the structure. We obtain a ZPL of 0.96 eV, well within the telecom region. The transition from the lower orbital ($d_{xy} + d_{xz}$) to $d_{x^2-y^2}$ is significantly higher with a ZPL of 1.18 eV (and a TDM of 3.0 D). The first excitation with ZPL of 0.96 eV results in a $\Delta Q$ of 2.47, Huang-Rhys factor of 8.66, and overall results in a Debye-Waller factor of 0.017 %. Similar values have been reported by Li and coauthors on C$_S$ in WS$_2$ (0.003%)[33]. On the other hand, the second excitation of Co$_S$ with a ZPL of 1.18 eV and a transition dipole moment 3.0 D exhibits a Debye-Waller factor of around 30%. A photonic cavity may be required to significantly enhance the zero-phonon emission[15,48]. All these results confirm the interest of the neutral Co substitutional defect as it combines emission in the telecom and possesses doublet spin multiplicity.

## Co$_S$ fabrication and characterization

In order to benchmark the presented screening approach, we create and characterize the Co$_S$ defect. Comparisons between the specific energy levels and effective orbital symmetries enable a direct comparison with the HT screening approach and first-principles

computations in general. In order to fabricate the Co$_S$ defect in WS$_2$, we make use of a tailored experimental workflow inside a low-temperature and ultrahigh vacuum (UHV) scanning probe microscope (SPM) that is shown in Fig. 4a–c. Sulfur vacancies (V$_S$) within otherwise as-grown WS$_2$ are created by resistively heating the sample and, in tandem, exposing it to a low incidence angle Ar$^+$ sputtering beam (Fig. 4a)[49]. This technique produces a high density of V$_S$ available for functionalization and subsequent reactivity. As adsorbed cobalt has been shown to be unstable on pristine TMD systems, such as MoS$_2$ and WS$_2$, we are able to make use of adsorbed instability near the VBM of WS$_2$ (below −1.3 V) to systemically induce diffusion and/or evaporation events with the SPM tip[50,51]. A Co physical vapor deposition apparatus (Fig. 4b) in UHV deposits randomly adsorbed Co to a defective WS$_2$/MLG/SiC(0001) sample, which is held at liquid helium temperatures, at submonolayer coverage. The bias over an adsorbed Co atom can then be ramped towards the tip-induced diffusion energy range to effectively excite the Co adatom into a V$_S$ for Co$_S$ defect creation (see Supplementary Fig. 6 for adsorbate behavior on as-grown WS$_2$). Figure 4d–g shows the resulting data at each fabrication workflow step with scanning tunneling micrographs. Linear defects are identified as one-dimensional inversion domains, which are a result of the V$_S$ creation process and have been described in detail elsewhere[49]. We then focus on the realization of Co$_S$. STM images, taken in constant-current mode, over a single Co defect are acquired before a Co diffusion event and after Co$_S$ formation, where the apparent height is reduced by 0.14 nm (Fig. 4h).

In order to investigate the evolved electronic structure with SPM, we make use of STS and differential conductance mapping, which are representative of the local density of states (LDOS) over a given defect. Point STS over Co$_S$ is shown in Fig. 5a, b, where in-gap states near 0.36 eV and 0.47 eV are measured. To make a clear distinction between adsorbed Co states, V$_S$, as-grown WS$_2$, and Co$_S$, point spectra are compared in Supplementary Fig. 7. We attribute peak broadening to electronic–phonon coupling, where effective electron–phonon coupling strength is estimated with a single-mode Franck-Condon model[16]. We include multiple phonon modes and additional quanta of each mode (available for co-excitation) in the description detailed in

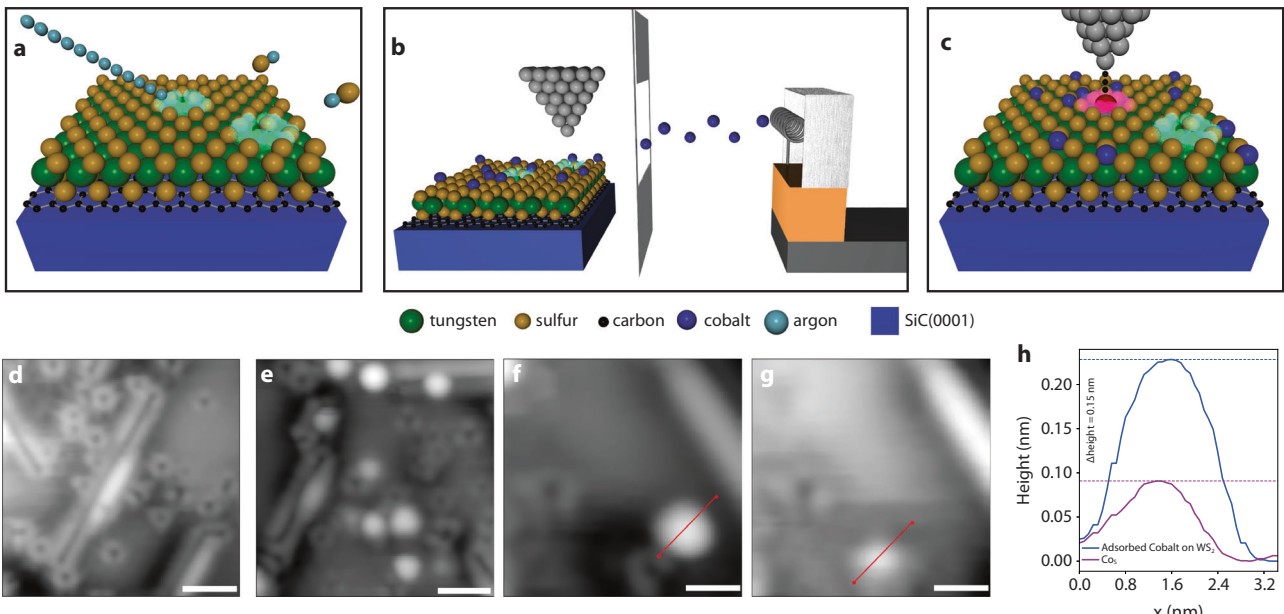

● tungsten  ● sulfur  ● carbon  ● cobalt  ● argon  ■ SiC(0001)

**Fig. 4 | Co$_S$ defect formation and characterization. a** The process of forming a high density of V$_S$, (**b**) low-temperature deposition of Co atoms in situ, and (**c**) subsequent placement into a sulfur vacancy (V$_S$) with the assistance of the STM probe that is used to selectively manipulate atoms at voltage ranges below −1.3 V is shown schematically. Corresponding scanning tunneling micrographs that capture WS$_2$/Gr/SiC(0001) (**d**) after defect introduction via Ar$^+$ bombardment and **e** post Co-deposition are plotted ($I_{tunnel}$ = 30 pA, $V_{sample}$ = 1.2 V). Scale bars, 2 nm. STM

images (**f**) before a voltage excitation and (**g**) after Co substitution within an identified V$_S$ are also shown ($I_{tunnel}$ = 30 pA, $V_{sample}$ = 1.2 V, $V_{excitation}$ = −2.1 V). Scale bars, 2 nm. $I_{tunnel}$ is the tunneling current, $V_{sample}$ is the sample bias voltage, and $V_{excitation}$ is the applied excitation voltage. **h** The apparent height difference of Co$_S$ compared to adsorption atop as-grown WS$_2$ is measured to be 0.15 nm, taken from line scans across both (**f**) (maxima shown with blue dashed line) and (**g**) (maxima shown with magenta dashed line) red highlighted regions.

Supplementary Fig. 8 to explain the dI/dV signal strength and broadening observed beyond the model approximation. Additionally, a resonance peak is identified at negative voltages (−0.84 ± 0.06 eV) that is attributed to electronic charging from the underlying substrate to Co$_S$, which shifts the defect to an anion state, where an electron is, on average, donated to available Co$_S$ defect levels. Spatially resolved DOS below the charging onset is comparable to that of the occupied orbitals in the anionic state and to the charge-neutral state above this onset. Figure 5c–e shows high-resolution differential conductance image maps that detail electronic orbital densities measured at −0.9 eV, 0.36 eV, and 0.47 eV. The LDOS at these energies are further benchmarked against calculations at the PBE0 level of theory and shown in Fig. 5f–h for each energy range experimentally measured, where Co$_S$ unoccupied orbitals are hybridized with bonded W atoms and are ~1.5 nm in diameter (see Supplementary Fig. 9 for simulated STS for charge states presented). We find strong agreement between experiment and theoretically obtained energy levels and orbital symmetries, where we can then assign the dI/dV peak at 0.36 eV to predominately $d_{x^2-y^2}$ orbital density, and the peak measured at 0.47 eV to a mixing of $d_{yz}$ and $d_{xz}$ orbitals at the Co$_S$ charge neutral state. The peak at −0.84 eV is attributed to the Co$_S$ charging (to Co$_S^{-1}$), and is discussed in further detail below. Quantitatively, the $d_{x^2-y^2}$ state is experimentally 0.64 eV below the CBM while theory predicts a level 0.5 eV below the CBM, indicating a good agreement.

We attribute the sharp peak at −0.84 eV to a charging process of the neutral cobalt to the anionic Co$_S^{-1}$ state. This charging is due to the localized tip-induced band bending process that has been described in the literature on similar systems[16,52]. The Co$_S$ lowest unoccupied state is occupied at adequate negative voltages and alters the Co$_S$ charge state making it anionic, as detailed in Fig. 5i, j. The E$_F$ of WS$_2$ has been shown to be driven by the heterostructure with graphene[53], where graphene is more susceptible to local doping and, here, is altered so that an electron is on average donated to the Co$_S$ defect. The neutral/anionic charge transition level is computed to be around 2.1 eV above

the VBM (see Fig. 3) which is close to the charge transition level for V$_S$ (see Supplementary Fig. 10) for which a charging peak at a similar position is observed for the same type of sample[16]. The charging peak near −0.84 eV varies spatially as the bias is ramped to more negative values: the radius of the ring around the defect center increases. In order to increase STS statistics, we perform an autonomous hyperspectral experiment over Co$_S$ (see Supplementary Notes 1 and 2 in addition to Supplementary Figs. 11–14)[21]. The charging peak is found to energetically shift between a minimum of −0.924 eV and a maximum of −0.627 eV during point STS measurements, which amounts to a ~0.3 eV tip-induced bending range of available states. This is near the 0.3 eV onset of the measured lowest unoccupied state, with a peak position of 0.36 eV, that is above the E$_F$ (as shown in Fig. 5a), enabling Co$_S$ to behave as an electron acceptor. Spatially-resolved charging ring formation as a function of applied bias is shown in Supplementary Fig. 15, where line scans taken across differential conductance maps from the defect center to outside the charging region highlight a shift to larger distances at more negative voltages. Outside the defect charging region, the substrate remains in a neutral state, which is verified with STS around pristine WS$_2$ regions (see Supplementary Fig. 16 for additional differential conductance mapping). While the charging process makes the identification of states closer to the VBM less straightforward, we note that, inside the charging ring, a state around the cobalt is observed. This state has the form of a $d_{z^2}$ orbital as expected from the computed LDOS of the neutral cobalt defect in that energy range (Fig. 5c and Supplementary Fig. 9b). The better comparison is with the LDOS of the Co$_S^{-1}$ as the defect should be charged within the ring. Theory predicts a reorganization of orbitals, an upward shift of the $d_{z^2}$ and a change of symmetry going from C$_s$ to C$_{3v}$ when Co$_S$ becomes negatively charged (see Supplementary Fig. 5). From this picture, we expect the $d_{z^2}$ state for Co$_S^{-1}$ to be 1 eV lower than the $d_{x^2-y^2}$ state from Co$_S^0$. We found experimentally a value of 1.26 eV. If there is an upward shift of $d_{z^2}$ when charged, it is smaller in the experiment than in theory. This discrepancy could come from the influence of the

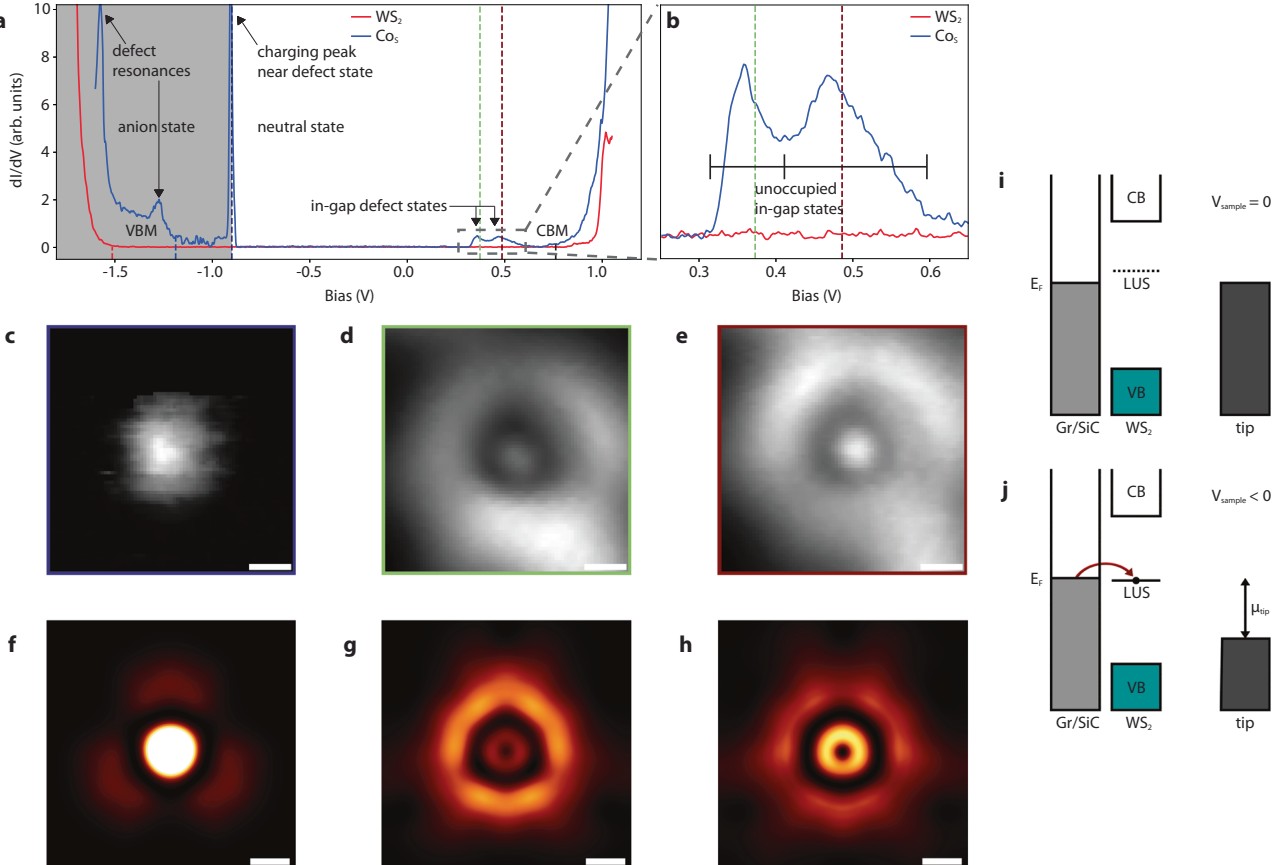

**Fig. 5 | Experimental and simulated Co$_S$ scanning tunneling spectroscopy.**
**a** Scanning tunneling spectra (STS) recorded on a Co$_S$ defect and the as-grown WS$_2$ monolayer on graphene ($V_{modulation}$ = 5 mV), where defect resonances, VBM and CBM onsets, in-gap states, and the shift between neutral (white background) to an anionic charge state (gray background) are labeled. $V_{modulation}$ is the bias modulation. **b** In-gap states identified are located at peak maxima of 0.36 eV and 0.47 eV, each with a full-width half maximum near 0.045 eV. Peak widths are broadened due to vibronic excitations (black lines). Differential conductance (dI/dV) imaging maps over the defect are depicted at (**c**) −0.9 eV (vertical black dashed line in **a**), (**d**) 0.373 eV (vertical green dashed lines in **a, b**), and (**e**) 0.486 eV (vertical red dashed

lines in **a, b**) ($V_{modulation}$ = 5 mV), showing Co$_S$ orbital geometries. Scale bars, 0.25 nm. **f**–**h** Simulated STS maps using PBE0 over Co$_S$ orbitals identifying energy range densities near experimentally measured values. Scale bars, 0.25 nm. Iso-contour value, $7 \times 10^{-6}$ Å$^{-3}$. A charging peak is identified in (**a**), where the (**i**) lowest unoccupied Co$_S^0$ state becomes (**j**) resonant with the E$_F$ of the substrate and an electron is donated to the lowest unoccupied state (LUS) at sufficient $V_{sample}$ (or equivalent tip potential, $\mu_{tip}$) produce the Co$_S^{-1}$ defect. Both (**c**) and (**f**) are representative of the Co$_S^{-1}$ orbital densities collected at the specified energy (the charging ring onset in (**c**) is removed for clarity).

---

dielectric environment of the graphene/SiC contacts that is not modeled in our WS$_2$ system in vacuum. In any case, next to the $d_{x^2-y^2}$, $d_{yz}$ and $d_{xz}$ Co state within the band gap, an additional Co $d_{z^2}$ state is observed within the band gap (and 1.26 eV lower than the $d_{x^2-y^2}$ state) confirming the theoretical results that Co in WS$_2$ can lead to a two-level system of great interest as a OQD.

## Discussion

We use HT computational screening to search for promising quantum defects in WS$_2$. Based on a database gathering computed properties for 757 charged defects in WS$_2$, we identify a handful of promising quantum defects with high brightness and in-gap defect states compatible with optical emission in the telecom or visible range. We fabricate the Co$_S^0$ defect, which we anticipate to exhibit brightness, a spin-doublet ground state, and a computed ZPL in the telecom at 0.966 eV[29,33,54], through metal deposition and subsequent sulfur vacancy substitution by cobalt with an STM tip. STM and STS analysis indicates cobalt-related defect states within the band gap confirming the computational prediction and the interest of Co$_S$ as an OQD.

Our HT data indicates that fundamental electronic structure reasons make transition metal substitution on sulfur sites more likely to lead to a OQD with in-gap defect states that could emit in the telecom

or visible than for the tungsten substitution. This motivates more efforts in the community along that direction. The fabrication process and HT computational screening used to identify Co$_S$ highlight the capability of combining HT screening and advanced synthesis techniques to identify and realize promising OQDs. This can be performed across a wide range of atomic species within 2D materials and other hosts with many yet to be experimentally realized, which can be executed for a number of different desired material properties, e.g., from catalysis to QIS.

## Methods

### Scanning probe microscopy (SPM) measurements

All measurements were performed with a Createc GmbH scanning probe microscope operating under ultrahigh vacuum (pressure < $2 \times 10^{-10}$ mbar) at liquid helium temperatures ($T$ < 6 K). Either etched tungsten or platinum-iridium tips were used during the acquisition. Tip apexes were further shaped by indentations onto a gold substrate for subsequent measurements taken over a defective substrate. STM images are taken in constant-current mode with a bias applied to the sample. STS measurements were recorded using a lock-in amplifier with a resonance frequency of 683 Hz and a modulation amplitude of 5 mV. Band gaps from STS were determined by applying a

linear fit to both the valence and conduction band edge, and the bottom of the band gap in log(dI/dV)[55].

## Sample preparation

Monolayer islands of $WS_2$ were grown on graphene/SiC substrates with an ambient pressure CVD approach (See Supplementary Fig. 17). A graphene/SiC substrate with 10 mg of $WO_3$ powder on top was placed at the center of a quartz tube, and 400 mg of sulfur powder was placed upstream. The furnace was heated to 900 °C and the sulfur powder was heated to 250 °C using a heating belt during synthesis. A carrier gas for process throughput was used (Ar gas at 100 sccm) and the growth time was 60 min. The CVD-grown $WS_2$/MLG/SiC was further annealed in vacuo at 400 °C for 2 h. $WS_2$ was sputtered with an argon ion gun (SPECS, IQE 11/35) that operated at 0.1 keV energy with 60° off-normal incidence at a pressure of $5 \times 10^{-6}$ mbar and held at 600 °C. A rough measure of current ($0.6 \times 10^{-6}$ A) enabled the argon ion flux to be estimated at ($1.5 \times 10^{-13}$ ions/cm²s), where the sample was irradiated for up to 30 seconds. Cobalt was deposited at a pressure of $1 \times 10^{-9}$ mbar for 60 s with the sample held at 5 K.

## Neural network and Gaussian process implementation

The acquisition software used for autonomous experimentation was gpSTS, which is a library for autonomous experimentation for scanning probe microscopy[21,56]. An Intel Xeon E5-2623 v3 CPU with 8 cores and 64 GB of memory combined with a Tesla K80 with 4992 CUDA cores was used for training the neural network. Training data for $WS_2$ and $V_S$ was combined with $Co_S$ spectra obtained from an extended autonomous run.

## First-principles calculations

We considered 57 elements that could substitute for W and S in the construction of a $WS_2$ quantum defect database, as highlighted in the periodic table in Supplementary Fig. 1. This collection covers the majority of the elements except the rare-earth elements and noble gases. All defect computations were performed at DFT level using automatic defect workflows that are implemented in ATOMATE software package[57–59]. The defect structure generations and the formation energy computations are performed using PYCDT. The DFT calculations were performed using Vienna Ab-initio Simulation Package (VASP)[60,61] and the projector-augmented wave method[62] with the Perdew-Burke-Ernzerhof (PBE) functional[63]. Each charged defect is simulated in a 144-atom orthorhombic supercell and with a vacuum of ~14 Å. A plane-wave basis energy cutoff of 520 eV was used and the Brillouin zone is sampled using Γ point only. The defect structures were optimized at a fixed volume until the forces on the ions were smaller than 0.01 eV/Å. The charge states of each defect are determined by considering all the oxidation states of the elements documented in the ICSD database[58] and taking into account the formal charges in $WS_2$ ($W^{4+}$ and $S^{2-}$). The total energy of the charged defects was further corrected to overcome the finite-size effect using the method of refs. [64,65] as implemented in SLABCC[66].

The above procedures generated overall 757 substitutional charged defects in monolayer $WS_2$. Based on the defect formation energy, we first identified 260 charged defects that are thermo-dynamically stable, meaning their charge states are accessible in a certain $E_F$ range. Of these, 89 defects exhibit singlet ground states, 94 show doublet character, 48 are triplets, and 16 are in higher states. Among these stable defects, we further search for the ones that possess two in-gap, localized levels that would enable the optical intra-defect transition. The localization is defined using inverse participation ratio (IPR) as detailed below. We considered levels with IPR larger than 0.05 as localized states

(bulk-like states in general have IPR smaller than 0.01 in $WS_2$). This trimmed down the list to 143 candidates, among which 112 have non-singlet ground states. The classification of singlets and multiplets is based on the electronic structure of the defect. In this case, the singlets and multiplets refer to the total magnetic quantum number of the unpaired electrons. Thus, defects with all electrons paired are classified as singlet, while those with one or two paired electrons are classified as doublet, triplet, etc. We note that due to limitations of Kohn-Sham (KS) DFT and the possibility of spin contamination for spin-polarized systems, more powerful methods such as spin-flip Bethe-Salpeter are required in general to rigorously determine the total spin $S$[67]. Finally, we screened out the ones that would emit at telecom wavelength with reasonable brightness. The emission wavelength is approximated using the single-particle KS energy difference using the single-shot PBE0 incorporating 7% of Fock exchange. We refrained from applying potential corrections at this stage as the KS energy difference is largely unaffected by the electrostatic finite-size effect. The brightness of the optical transition is approximated by the transition dipole moment (TDM) as detailed below. To search for the most relevant transitions, we consider the transitions that give the smallest energy difference, while also allowing an energy window of up to 100 meV to take into account the errors and band degeneracy. We then identified the transition with the largest TDM as the most relevant transition. The above procedures recommend 17 non-singlet candidates that emit at least 750 meV with a TDM of 3 D, as shown in Supplementary Table 1.

The localization of an orbital is described using the IPR. For a given KS State, the IPR is evaluated based on the probabilities of finding an electron with an energy $E_i$ close to an atomic site $\alpha$[68–70]:

$$\chi(E_i) = \frac{\sum_\alpha \rho_\alpha^2(E_i)}{\left[\sum_\alpha \rho_\alpha(E_i)\right]^2}, \quad (1)$$

where the summation runs over all atomic sites $\alpha$. The participation ratio $\chi^{-1}$ stands for the number of atomic sites that confine the wave function. Thus, a larger (smaller) IPR indicate a localized (delocalized) state. IPR is unitless ranging between 0 and 1. We computed IPR using VASP PROCAR. The optical transition dipole moment was evaluated by the PYVASPWFC code based on the single-particle wavefunction calculated at the PBE level[71]. The transition dipole moment is written as:

$$\boldsymbol{\mu}_k = \frac{i\hbar}{(\epsilon_{f,k} - \epsilon_{i,k})m} \langle \psi_{f,k} | \mathbf{p} | \psi_{i,k} \rangle, \quad (2)$$

where $\hbar$ is the Planck constant, $\epsilon_{i,k}$ and $\epsilon_{f,k}$ are the eigenvalues of the initial and final states, $m$ is the electron mass, $\psi_i$ and $\psi_f$ are the initial and final wavefunctions, and $\mathbf{p}$ is the momentum operator.

For selected substitutional defects, we carried out the fully self-consistent hybrid functional (PBE0) calculations including structural relaxations. In line with the single-shot PBE0 calculations and following previous work[37], we described the defect levels using the mixing parameter $\alpha = 0.07$ for the Fock exchange, which generally satisfies the Koopmans' condition for localized defects in monolayer $WS_2$. On the other hand, we used $\alpha = 0.22$ for the pristine $WS_2$ to determine the band-edge position. The alignment of defect levels with respect to the band edges was then achieved through the vacuum level which serves a common reference level. Spin-orbit coupling is taken into account unless otherwise specified. We used a planewave cutoff energy of 400 eV and a $2 \times 2 \times 1$ **k**-point mesh for ground-state calculations. The zero-phonon line was assessed using a single Γ point by imposing occupation constraints (constrained DFT[28]). For charged defects, the total energies are subject to finite-size effects and were corrected by the method of refs. [64,65] as implemented in SLABCC[66], whereas the single-particle KS levels were corrected by the potential correction

scheme of ref. 72. The simulated STM images were plotted at a constant height of 3.5 Å above the surface using the STM-2DScan package[73] based on the Tersoff-Hamann theory[74].

## Data availability

The computational dataset used in this work has been made publicly available at https://defectgenome.org. Additional data that support the findings of this study are available from the corresponding authors on request.

## Code availability

The code used for the findings of this study is available from the corresponding authors on request.

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

## Acknowledgements

This work was supported by the U.S. Department of Energy, Office of Science, Basic Energy Sciences in Quantum Information Science under Award Number DE-SC0022289. This work was supported as part of the Center for Novel Pathways to Quantum Coherence in Materials, an Energy Frontier Research Center funded by the U.S. Department of Energy, Office of Science, Basic Energy Sciences. Work was performed at the Molecular Foundry and at the Advanced Light Source supported by the Office of Science, Office of Basic Energy Sciences, of the U.S. Department of Energy under contract no. DE-AC02-05CH11231. S.K. and J.A.R. acknowledge support from the National Science Foundation Division of Materials Research (NSF-DMR) under awards 2002651 and 2011839. N.K. and D.A.S. acknowledge support from the National Science Foundation award DMR-2144317, and the Merced nAnomaterials Center for Energy and Sensing (MACES), a NASA-funded research and education center, under award NNH18ZHA008CMIROG6R. B.A.B. was supported by the U.S. Department of Energy, Office of Science, Basic Energy Sciences, CTC and CPIMS Programs, under Award DE-SC0019053. This research used resources of the National Energy Research Scientific Computing Center, a DOE Office of Science User Facility supported by the Office of Science of the U.S. Department of Energy under Contract No. DE-AC02-05CH11231 using NERSC award BES-ERCAP0020966. Additional computational resources were provided by the Multi-Environment Computer for Exploration and Discovery (MERCED) cluster at UC Merced, funded by National Science Foundation Grant no. ACI-1429783.

## Author contributions

G.H., A.W.-B., J.C.T., S.G. and A.R. (Archana Raja) conceived the overall project. W.C. (Wei Chen), Y.X., B.A.B., W.C. (Weiru Chen), N.K., S.G., D.A.S., G.-M.R. and G.H. performed the theoretical simulations along with compiling the database for quantum defect search. J.C.T., J.Z., and A.R. (Antonio Rossi) performed STM/STS experiments and subsequent analysis with support from E.S.B., A.S., D.F.O., E.R., A.R. (Archana Raja), and A.W.-B. J.C.T. and M.M.N. implemented autonomous experimentation. Z.Y., D.Z., S.K., J.A.R. and M.T. carried out sample growth. All authors discussed the results and contributed towards the manuscript.

## Competing interests

The authors declare no competing interests.

## Additional information

¹Molecular Foundry, Lawrence Berkeley National Laboratory, Berkeley, CA 94720, USA. ²Materials Sciences Division, Lawrence Berkeley National Laboratory, Berkeley, CA, USA. ³Thayer School of Engineering, Dartmouth College, Hanover, NH 03755, USA. ⁴Institute of Condensed Matter and Nanoscicence, Université Catholique de Louvain, Louvain-la-Neuve 1348, Belgium. ⁵Department of Physics, University of California, Merced, Merced, CA 95343, USA. ⁶Advanced Light Source, Lawrence Berkeley National Laboratory, Berkeley, CA 94720, USA. ⁷Department of Materials Science and Engineering, The Pennsylvania State University, University Park, PA 16082, USA. ⁸Center for Two-Dimensional and Layered Materials, The Pennsylvania State University, University Park, PA 16802, USA. ⁹Department of Physics, The Pennsylvania State University, University Park, PA 16802, USA. ¹⁰Department of Chemistry, The Pennsylvania State University, University Park, PA 16802, USA. ¹¹Applied Mathematics and Computational Research Division, Lawrence Berkeley National Laboratory, Berkeley, CA 94720, USA. ¹²These authors contributed equally: John C. Thomas, Wei Chen, Yihuang Xiong. ✉e-mail: jthomas@lbl.gov; afweber-bargioni@lbl.gov; geoffroy.hautier@dartmouth.edu

