## [Peer Review File · Nature Communications]

A substitutional quantum defect in WS_2 discovered by high-throughput computational screening and fabricated by site-selective STM manipulationREVIEWER COMMENTS

Reviewer #1 (Remarks to the Author):

The authors report on high throughput screening calculations for various metallic dopants in WS₂ and the fabrication and characterization of one of the most promising dopants (Co) at the atomic scale with UHV STM. Optically-active defects with spin structure represent extremely promising components of quantum information processing, communications and sensing platforms, and dopants in 2D materials present an excellent path to realize a wide array of dopant/host combinations. High throughput screening calculations that determine electronic structure, spin structure, and oscillator strength are a critical first step, and the ability to fabricate and characterize them at the atomic scale provides not only important feedback on the calculations, but a route to physically realizing these systems. While I found some of the discussion on the experimental analysis confusing, I find the calculations and the experiments compelling and recommend publication after several issues are clarified.

1. While I think the focus on a single TMDC is reasonable for the scope of the paper, what motivates WS₂ rather than a different TMDC?
2. Pg. 5: For the statement “while substitutional transition metals on W sites are easy to synthesize, ...” would benefit from a reference.
3. Fig. 3b: I cannot see a difference between the ‘empty’ and ‘filled’ rectangles?
4. Pg. 7: Question: The authors calculate the ZPL with “constrained DFT by imposing the occupation of the unoccupied $d_{x^2-y^2}$ state and relaxing the structure.” Do they also need to force the d_{z^2} state to be unoccupied?
5. Pg. 7: Also, the authors state “Transition from the d_{z^2} to the next orbital (d_{xy})...” do the authors mean d_{yz} , d_{xz} rather than d_{xy} ? In Fig. 3b, d_{xy} is below and close to d_{z^2} . If not, please clarify.
6. Does the CVD method employed to grow WS₂ on graphene/SiC provide only monolayer thickness? Or multiple layers? A larger scale image of the WS₂ on G/SiC and a cross-section with the step height shown would help clarify the monolayer thickness of the WS₂ (in the paper or SI).
7. On the Co_S dopant generation, the authors state “The bias over an adsorbed Co atom can then be ramped towards the tip-induced diffusion energy range to effectively excite the Co adatom into a VS for CoS defect creation.” The authors should provide more details (possibly in the SI) on how they drove adsorbed Co atoms into S vacancy sites and generate a bound dopant atom, or provide a reference where this is described in detail.
8. Bias is applied to the sample? Please clarify if not already stated.
9. Fig. 5: How do the authors know that the two states observed at 0.36V and 0.47V are closer to the CB rather than the VB? With TIBB, isn't it possible they are closer to the VB and are being pulled above the Fermi level and into the bias window at positive sample bias?
10. Fig. 5: The STS charging peak is identified with the pulling the lowest unoccupied state (LUS?) down to the Fermi level through tip-induced band-bending (TIBB). On Pg. 12, are the authors equating this with the state characterized by the STS peak at 0.36 eV?
11. I do not understand the discussion on Pg. 12 as to how the authors arrived at a 0.3 eV value for the TIBB? This will be voltage dependent ... are they only referring to the LUS charging peak? Was this based

on the supplemental Fig. 8 and autonomous STS? Can't this be quantified more systematically?

12. Additionally, the TIBB will distort all of the apparent energies and widths of the peaks in STS when comparing the measurements to theory; are the authors accounting for this when they compare these measurements to theory?

13. Pg. 12: I am confused by the discussion of the anionic state. Is this the state identified in Fig. 5a? The authors state that they found it to be lower in energy from the $d_{x^2-y^2}$ by 1.3eV ... where do I see this in the data?

14. Fig. 5: The scale bars in c-e look slightly different in size from the scale bars in f-h, despite the fact they are both supposed to be 0.25 nm. The authors should confirm the theory vs experiment comparison is showing the same field-of-view.

Reviewer #2 (Remarks to the Author):

In this work, John C. Thomas and co-workers conducted a high-throughput first-principles computational analysis to explore potential quantum defects within WS₂. Through this analysis, they pinpointed neutral cobalt substitution for sulfur (CoS) as a promising quantum defect, featuring localized energy levels conducive to bright telecom emission. Subsequently, the authors fabricated CoS defect in WS₂ via a three-step process: 1. Employing gentle Ar⁺ sputtering to create a sulfur vacancy on the WS₂ surface; 2. Depositing Co atoms onto the defective WS₂ surface under ultra-high vacuum (UHV) conditions and at liquid helium temperatures; 3. Utilizing tip-induced migration to activate Co adatoms within a sulfur vacancy. Employing STM and STS measurements, the authors demonstrated in-gap defect states arising from d orbitals, along with charging peaks attributed to tip-induced band bending. These findings were further corroborated by theoretical calculations. The findings from searching potential quantum defects via calculations to the realization and characterization of the targeted defect using STM/STS are remarkable. I highly recommend the publication of this work in Nature Communications upon addressing the following outlined issues.

1. It would be beneficial to include specific metrics or references highlighting the resource and time-saving advantages of employing high-throughput first-principles computational screening for potential quantum defects compared to traditional methods?
2. In page 4, line 12, the authors claim that "We only consider defects with a charge state that is stable within a certain Fermi level and with transitions between defect levels localized within the band gap.". What is the defect density used for the calculations? Will the defect density affect the position of Fermi level and change the charge state of the calculated defects?
3. In page 4, line 21, the authors claim that spin multiplets are of greater technological interest. Please explain this point more specifically with reference.
4. It would be beneficial to explain how lower formation energies may indicate greater stability and why this criterion is significant in assessing promising defect candidates.
5. All the defects mentioned in the main text should be labelled in Fig.1 b (e.g. TiS).
6. Although Fig. 4h shows the height difference for the Co adatom before and after tip-induced migration, such difference can still be assigned to different adsorption site (e.g. from on the sulfur atom

- to between the sulfur atoms). Any experimental evidence (e.g. atom-resolved STM image of Fig. 4f and 4g) that can be used to determine the Co adatom adsorption site?
7. The width of the domain boundary in Fig. 4d and 4e is close to each other. Why the length of scale bar shows such a large difference?
 8. In Fig. 5a, the STS taken over CoS shows a sharp peak below -1.5V, what is the origin of this peak? Another charging peak?
 9. The combination of STM with AFM can better decipher the nature of defects (Nat. Nanotechnol 10.1038/s41565-023-01495-z, 2023, Phys. Rev. Lett. 128, 176801, 2022). The author may consider to discuss this in the manuscript.
 10. The simulations (Fig. 5g and 5h) look upside down compared to the experimental results (Fig. 5d and 5e). How do the authors determine the configuration of W atoms that are next to the CoS?
 11. The authors claim that “the CoS defect shows brightness, a spin-doublet ground state” in the conclusion section. Any experimental results to reinforce the conclusions drawn ?

Reviewer #3 (Remarks to the Author):

The Authors use High-throughput screening approach to find bright optically active defects in WS₂. They find neutral CoS 0 (cobalt substitution to sulfur in WS₂) with a spin-doublet ground, as potentially interesting defect for optical applications, as this defect exhibit localized states within the band gap and has large transition dipole moment. The authors then use scanning tunnelling microscopy (STM) to fabricate this defect and establish the identity of generated defect through comparison of STM results with first principles calculations. The work can of interest to the readers of Nature comm, after following revisions.

1. CoS 0 has a doublet spin ground state, so it's very unlikely (though possible) that it can be used for applications that exploit ground state spin-manipulation e.g. ODMR related applications. My question would be why Doublet defect such as this is any more favourable for quantum applications, then the singlet defects that authors found in their computational work, but chose not to fabricate those singlet defects?
2. The authors are proposing CoS 0 for optical applications probably as for single photon emission in the telecom range etc., but they have not done any specific optical characterisation, e.g. photoluminescent spectra for this defect.
3. The authors have calculated the ZPL of the CoS 0, but they did not mention how broad would the optical emission from this defect look like. What are the ΔQ , Debye-waller and Huang-Rhys factors for this optical transition?

I would suppose from the difference in the values of single particle KS levels and ZPL, that this emission is very broad and not narrow. If that is the case how would authors propose this defect for single photon

emitting related applications?

Minor comments:

1. The authors should be more specific when referring to defects with non-zero spin ground states, e.g. both doublets and triplets have “non-singlet spin multiplicity”, so authors should use terms doublets or triplets.
2. On a similar note, it would help if authors specifically mention how many triplets did they find?

REVIEWER COMMENTS

Reviewer #1 (Remarks to the Author):

The authors report on high throughput screening calculations for various metallic dopants in WS₂ and the fabrication and characterization of one of the most promising dopants (Co) at the atomic scale with UHV STM. Optically-active defects with spin structure represent extremely promising components of quantum information processing, communications and sensing platforms, and dopants in 2D materials present an excellent path to realize a wide array of dopant/host combinations. High throughput screening calculations that determine electronic structure, spin structure, and oscillator strength are a critical first step, and the ability to fabricate and characterize them at the atomic scale provides not only important feedback on the calculations, but a route to physically realizing these systems. While I found some of the discussion on the experimental analysis confusing, I find the calculations and the experiments compelling and recommend publication after several issues are clarified.

1. While I think the focus on a single TMDC is reasonable for the scope of the paper, what motivates WS₂ rather than a different TMDC?

Thank you for raising this relevant point, WS₂ has been predicted to have long spin coherence time T_2 of ~11 ms, which is longer than other TMDC such as MoS₂ (~2.19 ms)^{1,2}, which is critical for quantum information science applications. In addition, we have extensive expertise in forming defects within WS₂, however, we do also plan to extend this to additional TMDCs in future studies. This is also addressed in the text in the following line:

WS₂, specifically, is a highly modifiable TMD that has been predicted to have long spin coherence times (T_2 of ~11 ms)^{1,2}.

2. Pg. 5: For the statement “while substitutional transition metals on W sites are easy to synthesize, . . .” would benefit from a reference.

Thank you for your comment. The following references have been added to address metal-site doping vs chalcogen-site doping in addition with updates to the below sentence from the main text.

While substitutional transition metals on W sites are easy to synthesize^{3,4}, our screening results show that this is not the most promising approach for OQD discovery.

3. Fig. 3b: I cannot see a difference between the ‘empty’ and ‘filled’ rectangles?

Thank you for pointing this out. Figure 3b has been updated to enhance the distinction between occupied and unoccupied defect levels.

Fig. 3: Thermodynamic charge transition levels and electronic structure of CoS. **a** Formation energy of CoS as a function of Fermi level for the neutral and the two charged states. The charge transition levels, i.e., (+/0) and (0/−), are referenced to the band-edge positions of pristine WS₂ as obtained with PBE0 incorporating 22% of Fock exchange PBE0(0.22). **b** Orbital diagram of the localized defect states for neutral CoS. Resonant states within the valence band and conduction band manifolds are not depicted. The characters of the localized states are indicated. The occupied (unoccupied) states are shown by the filled (empty) rectangles, the height of which indicates the degree of dispersion. The band-edge positions refer to those of the pristine WS₂ obtained with PBE0(0.22). Energies are referenced to the vacuum level. SOC is not taken into account for the localized defect states. **c** Top view of the charge density (in blue) for the three CoS₀ defect states as indicated in **b**. The isovalue is 0.001 e/Å³.

4. Pg. 7: Question: The authors calculate the ZPL with “constrained DFT by imposing the occupation of the unoccupied $d_{y^2-x^2}$ state and relaxing the structure.” Do they also need to force the d_{z^2} state to be unoccupied?

Indeed the d_{z^2} state is forced to be unoccupied by the constrained DFT. We have updated the text to better reflect this with the sentence below.

The zero-phonon lines (ZPL) associated with this transition are computed within the constrained-occupation DFT by imposing the electron occupation (or needed unoccupation) of the d_{z^2} and $d_{x^2-y^2}$ states and relaxing the structure.

5. Pg. 7: Also, the authors state “Transition from the d_{z^2} to the next orbital (d_{xy}). . .” do the authors mean d_{yz} , d_{xz} rather than d_{xy} ? In Fig. 3b, d_{xy} is below and close to d_{z^2} . If not, please clarify.

The second possible transition is in fact from the lower occupied state $d_{xy} + d_{xz}$ to $d_{x^2-y^2}$. This has been corrected in the revised manuscript. The higher unoccupied state ($d_{yz} + d_{xz}$) is more extended and hybridizes with the W-d states, and as such we do not consider the transition to this state. We thank the referee for spotting this error.

Transition from the lower orbital ($d_{xy} + d_{xz}$) to $d_{x^2-y^2}$ is significantly higher with a ZPL of 1.18 eV (and a TDM of 3.0 D).

6. Does the CVD method employed to grow WS₂ on graphene/SiC provide only monolayer thickness? Or multiple layers? A larger scale image of the WS₂ on G/SiC and a cross-section with the step height shown would help clarify the monolayer thickness of the WS₂ (in the paper or SI).

Thank you for your suggestion. The samples measured were of monolayer WS₂/Graphene/SiC. An updated figure showing apparent height difference and identification with STS is now included in the supplementary information and the materials and methods.

Supplementary Fig. 17: **Monolayer WS₂ Identification.** **a** Scanning tunneling image over a WS₂ monolayer edge resting on a graphene/SiC(0001) substrate ($I_{tunnel} = 30$ pA, $V_{sample} = 1.2$ V). Scale bar, 10 nm. **b** A height profile taken across the blue line depicted in **a**, where a height difference of ~ 0.5 nm is measured. Regions are further verified with **c** scanning tunneling spectroscopy over both grown WS₂ (red circle in **a**) and the graphene/SiC substrate (gray circle in **a**) ($V_{modulation} = 5$ mV, $I_{set} = 150$ pA). A band gap of 2.5 eV is measured for as-grown WS₂, and graphene exhibits expected canonical band structure.

Monolayer islands of WS₂ were grown on graphene/SiC substrates with an ambient pressure CVD approach (See Supplementary Fig. 17).

7. On the Co_S dopant generation, the authors state “The bias over an adsorbed Co atom can then be ramped towards the tip-induced diffusion energy range to effectively excite the Co adatom into a VS for CoS defect creation.” The authors should provide more details (possibly in the SI) on how they drove adsorbed Co atoms into S vacancy sites and generate a bound dopant atom, or provide a reference where this is described in detail.

We apologize we were not as clear about how this is formed. Adsorbed Co has been shown to be mobile in the literature over both WS₂ and MoS₂^{5,6}. Additionally, we show this in greater detail in updated Supplementary Fig. 6, where atoms are easily displaced (under the tip) at the energy ranges specified. As the adsorbed atoms can be evaporated/displaced, having a neighboring advantageous defect enables the absorption of the defect, which is verified with scanning tunneling spectroscopy and by imaging in Figs. 4 and 5. We, additionally, make this spectroscopic difference more clear in Supplementary Fig. 7, where the apparent band gap of adsorbed Co states (predicted to be either on a W, S, or Hollow site)^{7,8} is significantly smaller than the apparent band gap of either a V_S or the as-measured Co_S, which both fall in the ~ 2 eV range. Adsorbed, on the other hand, shows a band gap of 0.97 ± 0.27 eV and tends to diffuse and/or evaporate below -1.3 eV.

Supplementary Fig. 6: **Tip-induced evaporation/diffusion.** **a** An atomically sharp tip is rastered across a adsorbed Co defect site. **b, c** Scanning tunneling micrographs depicting pristine WS_2 with submonolayer Co atoms adsorbed before local diffusion/evaporation events ($I_{\text{tunnel}} = 30 \text{ pA}$, $V_{\text{sample}} = 1.2 \text{ V}$). Scale bars, 20 nm and 3 nm, respectively. **d** After scanning the local region in **c** at excitation voltage of -1.4 V, the majority of atoms are evaporated (or diffused to a defect capable of absorption (i.e., V_S)). **e, f** Scanning tunneling images of the same large-scale and excited region, where the majority of Co atoms remain that have not been exposed to tunneling-induced motion ($I_{\text{tunnel}} = 30 \text{ pA}$, $V_{\text{sample}} = 1.2 \text{ V}$). Scale bars, 20 nm and 3 nm. Predicted stable sites of adsorbed Co before tip-induced excitation include above a W site, S site, and a hollow site, where any remaining Co adatoms, after a tip-induced event, are expected to remain in a more stable W site^{7,8}.

The bias over an adsorbed Co atom can then be ramped towards the tip-induced diffusion energy range to effectively excite the Co adatom into a V_S for Co_S defect creation (see Supplementary Fig. 6 for adsorbate behavior on as-grown WS_2).

Supplementary Fig. 7: **Point STS Comparison.** dI/dV spectra recorded on as-grown WS_2 (red), adsorbed Co atop WS_2 (black), the CoS defect (blue), and a typical V_S (gray) are presented above ($V_{modulation} = 5$ mV). The as-measured energy gap of an adsorbed Co is 0.97 ± 0.27 eV and 2.0 ± 0.05 eV for CoS . Both WS_2 and V_S recorded energy gaps and point spectra match values that have been reported in the literature⁹.

In order to make a clear distinction between adsorbed Co states, V_S , as-grown WS_2 , and CoS , point spectra are compared in Supplementary Fig. 7.

8. Bias is applied to the sample? Please clarify if not already stated.

Yes, the bias is applied to the sample. All scanning tunneling data make use of the following notation (where I_{tunnel} and V_{sample} can vary from image to image to showcase different energy regimes): ($I_{tunnel} = 30$ pA, $V_{sample} = 1.2$ V). We also added this clarification to the methods as below in our materials and methods.

STM images are taken in constant-current mode with a bias applied to the sample. STS measurements were recorded using a lock-in amplifier with a resonance frequency of 683 Hz and a modulation amplitude of 5 mV to sample bias.

9. Fig. 5: How do the authors know that the two states observed at 0.36V and 0.47V are closer to the CB rather than the VB? With TIBB, isn't it possible they are closer to the VB and are being pulled above the Fermi level and into the bias window at positive sample bias?

Thank you for your comment. As the bias is applied to the sample, unoccupied states are measured at positive sample bias and occupied states are measured at negative sample bias in conventional scanning tunneling spectroscopy⁹, where the CBM is the onset of continuous tunneling in the positive sample bias regime and the VBM is the onset of continuous tunneling in the negative sample bias regime (as in-gap states exhibit signal but return to zero-tunneling after a given energy). Our calculations would suggest that they are unoccupied and above the Fermi level at a charge neutral state. We would expect them to be filled in a charge negative state, as indicated in Supplementary Fig. 9. If the states were closer to the VBM and being pulled above the Fermi level in the positive bias regime, the states would have also been measured spectroscopically at negative bias, which we did not record in Supplemental Fig. 16.

Additionally, this should be accompanied by a discharging peak in the positive bias sample regime (as the states would be occupied below the Fermi level). Our experimental and calculated results would indicate that the two states measured at 0.36 and 0.47 eV are unoccupied, above the Fermi level, and closer to the onset of the CBM at 0.86 eV versus the VBM at -1.19 eV. Lastly, we expect the heterostructuring with the underlying graphene layer to play a large role in the TIBB, where graphene is more susceptible to doping and drives the Fermi level of WS_2 ¹⁰. We have also added the text below, so that this is fully clarified:

The E_F of WS_2 has been shown to be driven by the heterostructuring with graphene¹⁰, where graphene is more susceptible to local doping and, here, is altered so that an electron is on average donated to the Co_S defect.

10. Fig. 5: The STS charging peak is identified with the pulling the lowest unoccupied state (LUS?) down to the Fermi level through tip-induced band-bending (TIBB). On Pg. 12, are the authors equating this with the state characterized by the STS peak at 0.36 eV?

Yes, the in-gap states follow similar behavior to a V_S as shown in Supplementary Fig. 10. The peak maximum reported is 0.36 eV, however the states are broadened as indicated in Supplementary Fig. 8. We have modified the text as below to help clarify this question.

The charging peak is found to energetically shift between a minimum of -0.924 eV and a maximum of -0.627 eV during point STS measurements, which amounts to a ~ 0.3 eV tip-induced bending range of available states. This is near the 0.3 eV onset of the measured lowest unoccupied state, with a peak position of 0.36 eV, that is above the E_F (as shown in Fig. 5a), enabling Co_S to behave as an electron acceptor.

11. I do not understand the discussion on Pg. 12 as to how the authors arrived at a 0.3 eV value for the TIBB? This will be voltage dependent . . . are they only referring to the LUS charging peak? Was this based on the supplemental Fig. 8 and autonomous STS? Can't this be quantified more systematically?

Thank you for your question. As the LUS is pulled below the Fermi level, and the onset of the peak begins to exhibit an increased LDOS at 0.3 eV (onset of the in-gap state measured with a peak position at 0.36 eV), we attributed this to be the estimated amount of TIBB involved in charging this unoccupied state (which becomes occupied once charged). We hope the above clarification solidifies the presentation of our findings.

12. Additionally, the TIBB will distort all of the apparent energies and widths of the peaks in STS when comparing the measurements to theory; are the authors accounting for this when they compare these measurements to theory?

The main effect of the TIBB is the charging at negative biases. However, it has little to no influence on the energetic position of the unoccupied states, which we employed as benchmark for the theory. The amount of TIBB effects on VBM and CBM onsets has been shown to be on the order of 0.02 - 0.04 eV in Ge¹¹, and we expect the level of TIBB effects on measured in-gap states to be small due to the heterostructuring with graphene. In fact, this is why groups such as Michael Crommie and others use graphene as an underlying layer, is that graphene drives the Fermi level position of the heterostacked TMD material and enables a way of tuning Fermi level position. Here, the underlying metallic graphene substrate drives the Fermi level position within WS_2 ¹², a larger dielectric, and properties of graphene, such as massless charge carriers and high mobility, enable increased susceptibility to local doping and subsequent Fermi level shifts¹⁰. Co_S , in fact, shows similar behavior to the V_S system⁹, where the Fermi level of the underlying graphene substrate is shifted with sufficient bias to enable charge occupation of the LUS. We hope the text added from comments 9 and 10 help better clarify this point.

13. Pg. 12: I am confused by the discussion of the anionic state. Is this the state identified in Fig. 5a? The authors state that they found it to be lower in energy from the $d_{y^2-x^2}$ by 1.3eV ... where do I see this in the data?

Thank you for the question. We measure a d_{z^2} orbital, that is convoluted by charging events, at -0.9 eV. The difference in energy from the peak position of the $d_{y^2-x^2}$ is 1.26 eV. We thank the reviewer for pointing out this error.

We found experimentally a value of 1.26 eV. If there is an upward shift of d_{z^2} when charged, it is smaller in experiment than in theory. This discrepancy could come from the influence of the dielectric environment of the graphene/SiC contacts that is not modeled in our WS_2 system in vacuum. In any case, next to the $d_{x^2-y^2}$, d_{yz} and d_{xz} Co state within the band gap, an additional Co d_{z^2} state is observed within the band gap (and 1.26 eV lower than the $d_{x^2-y^2}$ state) confirming the theoretical results that Co in WS_2 can lead to a two-level system of great interest as a QKD.

14. Fig. 5: The scale bars in c-e look slightly different in size from the scale bars in f-h, despite the fact they are both supposed to be 0.25 nm. The authors should confirm the theory vs experiment comparison is showing the same field-of-view.

Thank you for pointing this out. We noted that the experimental figures (c-e) appear to have larger field of view compared with the simulated ones (f-h) despite having the same scale bar. This is due to the use of different isocontour ranges for the orbital density. The goal of these figures is to highlight the geometric features of the orbitals between the experimental measurements and theoretical predictions. However, we appreciate reviewer's point on this and we have now reported the isocontour value used for the simulation to ensure reproducibility ($7 \times 10^{-6} \text{ \AA}^{-3}$). We have now modified the main text to reflect this.

Fig. 5: Experimental and Simulated Co_S Scanning Tunneling Spectroscopy. **a** STS spectra recorded on a Co_S defect and the as-grown WS₂ monolayer on graphene ($V_{modulation} = 5$ mV). **b** In-gap states identified are located at peak maxima of 0.36 eV and 0.47 eV, each with a full-width half maximum near 0.045 eV. Differential conductance (dI/dV) imaging maps over the defect are depicted at **c** -0.9 eV, **d** 0.373 eV, and **e** 0.486 eV ($V_{modulation} = 5$ mV), showing Co_S orbital geometries. Scale bars, 0.25 nm. **f-h** Simulated STS maps using PBE0 over Co_S orbitals identifying energy range densities near experimentally measured values. Scale bars, 0.25 nm. Isocontour value, $7 \times 10^{-6} \text{ \AA}^{-3}$. A charging peak is identified in **a**, where the **i** lowest unoccupied Co_S⁰ state becomes **j** resonant with the E_F of the substrate and an electron is donated to produce the Co_S⁻¹ defect. Both **c** and **f** are representative of the Co_S⁻¹ orbital densities collected at the specified energy (the charging ring onset in **c** is removed for clarity).

Reviewer #2 (Remarks to the Author):

In this work, John C. Thomas and co-workers conducted a high-throughput first-principles computational analysis to explore potential quantum defects within WS₂. Through this analysis, they pinpointed neutral cobalt substitution for sulfur (Co_S) as a promising quantum defect, featuring localized energy levels conducive to bright telecom emission. Subsequently, the authors fabricated Co_S defect in WS₂ via a three-step process: 1. Employing gentle Ar⁺ sputtering to create a sulfur vacancy on the WS₂ surface; 2. Depositing Co atoms onto the defective WS₂ surface under ultra-high vacuum (UHV) conditions and at liquid helium temperatures; 3. Utilizing tip-induced migration to activate Co adatoms within a sulfur vacancy. Employing STM and STS measurements, the authors demonstrated in-gap defect states arising from d orbitals, along with charging peaks attributed to tip-induced band bending. These findings were further corroborated by theoretical calculations. The findings from searching potential quantum defects via calculations to the realization and characterization of the targeted defect using STM/STS are

remarkable. I highly recommend the publication of this work in Nature Communications upon addressing the following outlined issues.

1. It would be beneficial to include specific metrics or references highlighting the resource and time-saving advantages of employing high-throughput first-principles computational screening for potential quantum defects compared to traditional methods?

Thank you for raising this point. Targeted synthesis and characterization of 2D quantum defects can be a challenging and labor-intensive process, as demonstrated in our work. Exploring a large chemical space of defects would render the process prohibitively difficult. On the other hand, high-throughput first-principles computations could dramatically accelerate this process. In our study, each 2D defect computation in our database takes on average about 2.79 hours with 64 CPU cores. We refer to a review article that provides an in-depth discussion on how high-throughput computation can help explore the chemical space of materials, in this case, the chemical space of defects in 2D materials. We have now added the relevant reference.

Here, we use first principles high-throughput (HT) computing to build a database of point defects in WS₂ considering all possible substitutional defects from 57 elements, aiming to accelerate the exploration of defect chemical space in WS₂¹³.

2. In page 4, line 12, the authors claim that “We only consider defects with a charge state that is stable within a certain Fermi level and with transitions between defect levels localized within the band gap.”. What is the defect density used for the calculations? Will the defect density affect the position of Fermi level and change the charge state of the calculated defects?

Thank you for raising this relevant point. In our computational work, we only consider one point defect in the dilute limit and in different charged states. In this context, we obtain the formation energy of a defect in a certain charged state for a given Fermi level. For a given defect, different charge states will be stable in different Fermi level ranges. This is what is typically computed from first principles computations and we can refer to references^{14–16} for more details. In practice, the Fermi level is set by other defects (intentional such as dopants or not intentional), by the underlying substrate, or by gating. For instance, the NV center in diamond is stable in the -1 state in a range of Fermi level from 2.8 eV to 5.2 eV (according to Weber et al.¹⁷ and with respect to the valence band minimum). Lower Fermi level will stabilize the NV in the zero charge state (which will have different properties). Within these plots, it is possible that a certain defect charge state is never stable in any Fermi level (within the band gap) (e.g., the sulfur vacancy +1 charge state is never stable in WS₂, the sulfur vacancy is either 0 or -1)^{18,19}. A charged defect that is never stable within a certain Fermi level will not be realizable and we exclude them from our study. We have clarified this in the text:

Point defects in semiconductors can have different charge states depending on the Fermi level (E_F). Certain charge states are not stable for any E_F within the band gap. While we do not study how a given E_F can be achieved (e.g., through doping or gating) we only consider defects in a charge state that is stable for a range of E_F within the band gap. In addition, we focus on charged defects with possible optical transitions between defect levels localized within the band gap.

3. In page 4, line 21, the authors claim that spin multiplets are of greater technological interest. Please explain this point more specifically with reference.

Thank you for raising this point. The search of this work focuses on quantum defects with "paramagnetic" electronic states^{17,20,21} inspired by defects with spin multiplets such as NV center or T center in silicon. Quantum technology such as quantum communication is enabled by the spin-photon interfaces, where in this case, a defect with spin and spin-dependent optical transitions. Such spin may be further entangled through photons. Thus, defects with ground state singlet are of great technological relevance to single

photon emitters, however, our search aims to find spin-photon centers for various quantum technologies. We now further clarify this point in the main text:

For quantum applications, defects that possess a nonzero spin are often desirable and are called spin-photon interfaces²⁰⁻²³. In WS₂, spin multiplet defects only appear through sulfur substitution: ...

4. It would be beneficial to explain how lower formation energies may indicate greater stability and why this criterion is significant in assessing promising defect candidates.

Thank you for raising this point. The only criteria we applied on formation energy is to ensure that there was some Fermi level that would stabilize the given charged defect. We agree with the reviewer that lower formation energy could be important for stabilizing defects and synthesizing certain types of defects, however, for synthesizing methods, such as the STM method applied in this work, or ion implantation, the processes are not fully driven by thermodynamics. We have now updated the main text to clarify this:

No criteria on the formation energy other than the need for the charged defect to have a range of E_F in which it is stable was applied in our screening.

5. All the defects mentioned in the main text should be labelled in Fig.1 b (e.g. TiS).

Thank you for raising this point. As detailed in our method section "High-Throughput Search", we focused on searching for defects with electronic structures that permit transition between two localized levels in the bandgap. All the defects that are shown in Fig.1 b fulfill this requirement. Fig.1 b in fact includes the defect Ti_S⁰ that the reviewer pointed out, with an excitation energy of around 1.8 eV and a TDM of around 4 D. A few other 3d transition metal defects that were discussed in terms of the chemical trend of the electronic structures, however do not possess two localized levels in the bandgap, thus are not labeled in Fig.1 b. Details of the electronic structures of these 3d transition metal defects can be found in Supplementary Fig. 4.

6. Although Fig. 4h shows the height difference for the Co adatom before and after tip-induced migration, such difference can still be assigned to different adsorption site (e.g. from on the sulfur atom to between the sulfur atoms). Any experimental evidence (e.g. atom-resolved STM image of Fig. 4f and 4g) that can be used to determine the Co adatom adsorption site?

We agree with the reviewer that more data is needed to confirm that the Cobalt is truly sitting in a sulfur vacancy. We provide this data in the revised version. We physisorbed the Co adatoms on a pristine WS₂ substrate. We have the specific spectroscopic signature of pristine WS₂, a sulfur vacancy, and a Co atom sitting on top of pristine WS₂. After manipulating the Co adatom into sulfur vacancy, we identified a defect that appeared completely different from the sulfur vacancy. Furthermore, it also appears different with a lower apparent height than a cobalt atom physisorbed on WS₂. In addition, the newly formed defect has distinctly modified spectroscopic signature to both a sulfur vacancy and adsorbed cobalt adatoms on the pristine substrate (shown in newly added Supplementary Fig. 7). Last, while cobalt adatoms can be easily manipulated with the STM tip (as shown in Supplementary Fig. 6), the newly formed defect at the exact position as the previous sulfur vacancy cannot be modified by Tip induced approaches. All this is described in detail in Figure 5 and additional supplementary information. This is verified microscopically and spectroscopically as shown in more detail in Figure 5 and also in the supplemental information. We also include additional imaging of adsorbed states before and after tip-induced excitation.

Supplementary Fig. 7: **Point STS Comparison.** dI/dV spectra recorded on as-grown WS₂ (red), adsorbed Co atop WS₂ (black), the Co_S defect (blue), and a typical V_S (gray) are presented above ($V_{modulation} = 5$ mV). The as-measured energy gap of an adsorbed Co is 0.97 ± 0.27 eV and 2.0 ± 0.05 eV for Co_S. Both WS₂ and V_S recorded energy gaps and point spectra match values that have been reported in the literature⁹.

Supplementary Fig. 6: **Tip-induced evaporation/diffusion.** **a** An atomically sharp tip is rastered across a adsorbed Co defect site. **b, c** Scanning tunneling micrographs depicting pristine WS₂ with submonolayer Co atoms adsorbed before local diffusion/evaporation events ($I_{tunnel} = 30$ pA, $V_{sample} = 1.2$ V). Scale bars, 20 nm and 3 nm, respectively. **d** After scanning the local region in **c** at excitation voltage of -1.4 V, the majority of atoms are evaporated (or diffused to a defect capable of absorption (i.e., V_S)). **e, f** Scanning tunneling images of the same large-scale and excited region, where the majority of Co atoms remain that have not been exposed to tunneling-induced motion ($I_{tunnel} = 30$ pA, $V_{sample} = 1.2$ V). Scale bars, 20 nm and 3 nm. Predicted stable sites of adsorbed Co before tip-induced excitation include above a W site, S site, and a hollow site, where any remaining Co adatoms, after a tip-induced event, are expected to remain in a more stable W site^{7,8}.

7. The width of the domain boundary in Fig. 4d and 4e is close to each other. Why the length of scale bar shows such a large difference?

Thank you for your careful review. Upon verification, the scale bar in Fig. 4e has been updated as below.

Fig. 4: Co_S Defect Formation and Characterization. **a** The process of forming a high density of V_S, **b** low-temperature deposition of Co atoms *in situ*, and **c** subsequent placement into a V_S with the assistance of the STM probe that is used to selectively manipulate atoms at voltage ranges below -1.3 V is shown schematically. Corresponding scanning tunneling micrographs that capture WS₂/Gr/SiC(0001) **d** after defect introduction via Ar⁺ bombardment and **e** post Co deposition are plotted ($I_{tunnel} = 30$ pA, $V_{sample} = 1.2$ V). Scale bars, 2 nm. STM images **f** before a voltage excitation and **g** after Co substitution within an identified V_S are also shown ($I_{tunnel} = 30$ pA, $V_{sample} = 1.2$ V, $V_{excitation} = -2.1$ V). Scale bars, 2 nm. **h** The apparent height difference of Co_S compared to adsorption atop as-grown WS₂ is measured to be 0.15 nm, taken from linescans across both **f** and **g** red highlighted regions.

8. In Fig. 5a, the STS taken over CoS shows a sharp peak below -1.5 V, what is the origin of this peak? Another charging peak?

We appreciate the question, as it was not fully addressed in our initial version of the manuscript. The onset of the VBM is measured below -1.18 eV, where additional defect resonances (driven by a mix of Co, S, and W orbitals within the VBM) are measured at -1.25 eV and -1.5 eV. Charging causes the VBM onset to shift upwards in energy, which has been reported in the literature for similar effects in V_S⁹. The only charging peak is measured at -0.84 eV. The charging of the defect has little to no influence on the energetic position of the unoccupied states, which we employed as benchmark for the theory. The amount of TIBB effects on VBM and CBM onsets has been shown to be on the order of $0.02 - 0.04$ eV in Ge¹¹, and we expect the level of TIBB effects on measured in-gap states to be small due to the heterostructuring with graphene. In fact, this is why groups such as Michael Crommie and others use graphene as an underlying layer, is that graphene drives the Fermi level position of the heterostacked TMD material and enables a way of tuning Fermi level position. Here, the underlying metallic graphene substrate drives the Fermi level position within WS₂¹², a larger dielectric, and properties of graphene, such as massless charge carriers and high mobility, enable increased susceptibility to local doping and subsequent Fermi level shifts¹⁰. The similarity between Co_S to the V_S system⁹ becomes apparent as the Fermi level of the underlying graphene substrate is shifted with sufficient bias to enable charge occupation of the LUS. Figure 5 has been updated to highlight defect resonances and onsets of the VBM and CBM. Additionally, we add a reference for fitting the VBM and CBM to the methods and methods.

Fig. 5: Experimental and Simulated Co_S Scanning Tunneling Spectroscopy. **a** STS spectra recorded on a Co_S defect and the as-grown WS₂ monolayer on graphene ($V_{\text{modulation}} = 5$ mV), where defect resonances, VBM and CBM onsets, in-gap states, and the shift between neutral to an anionic charge state are labeled. **b** In-gap states identified are located at peak maxima of 0.36 eV and 0.47 eV, each with a full-width half maximum near 0.045 eV. Differential conductance (dI/dV) imaging maps over the defect are depicted at **c** -0.9 eV, **d** 0.373 eV, and **e** 0.486 eV ($V_{\text{modulation}} = 5$ mV), showing Co_S orbital geometries. Scale bars, 0.25 nm. **f-h** Simulated STS maps using PBE0 over Co_S orbitals identifying energy range densities near experimentally measured values. Scale bars, 0.25 nm. Isocontour value, $7 \times 10^{-6} \text{ \AA}^{-3}$. A charging peak is identified in **a**, where the **i** lowest unoccupied Co_S⁰ state becomes **j** resonant with the E_F of the substrate and an electron is donated to produce the Co_S⁻¹ defect. Both **c** and **f** are representative of the Co_S⁻¹ orbital densities collected at the specified energy (the charging ring onset in **c** is removed for clarity).

Band gaps from STS were determined by applying a linear fit to both the valence and conduction band edge, and the bottom of the band gap in $\log(dI/dV)$ ²⁴.

9. The combination of STM with AFM can better decipher the nature of defects (Nat. Nanotechnol 10.1038/s41565-023-01495-z, 2023, Phys. Rev. Lett. 128, 176801,2022). The author may consider to discuss this in the manuscript.

Yes, we certainly agree the combination of both AFM and STM is ideal. Our group has substantial experience of directly correlating ncAFM and STM on defects within 2D materials. However, the big challenge we faced was that a functionalized tip with carbon monoxide strongly interacts with the Co adatoms. That is, that the CO is pulled off the tip to interact with the Co adatom. Without the carbon monoxide at the end of the tip, it was not possible to get high resolution ncAFM data to make the

comparison relevant. We did not want to present high-resolution Co noncontact AFM data due to the irreproducibility. Our group is currently working on other types tip functionalization that will enable such measurements. During our measurements, ncAFM tips were consistently changed or lost when approaching Co defects.

10. The simulations (Fig. 5g and 5h) look upside down compared to the experimental results (Fig.5d and 5e). How do the authors determine the configuration of W atoms that are next to the CoS?

Thank you for pointing this out. The simulation results were indeed rotated and have now been updated in the main text (as shown in Fig. 5 above). Our simulation results do reveal an increased local density of states along cobalt coordinated tungsten atom sulfur sites, which gives a three-fold symmetry. This comparison enables assignment of tungsten atom positions, as shown in Supplementary Fig. 9.

11. The authors claim that “the CoS defect shows brightness, a spin-doublet ground state” in the conclusion section. Any experimental results to reinforce the conclusions drawn ?

We based this sentence on computational results. Experimentally, this is very difficult to verify on the atomic scale as we created these quantum emitters at this scale, one defect at a time. As such, the optical characterization is complicated for a few reasons, 1: the way we have functionalized a defect with cobalt is one defect at a time, which we are not able to easily read out optically (we would have to find the exact same position to do a measurement on a single Co defect, and finding the exact spot where we performed STM manipulation is inherently difficult). 2: if you pull these samples out to do any optical measurements, it needs to be pulled into ambient condition and could oxidize. For these two reasons, it is difficult. Additionally, doing these types of measurements in STM with tip-assisted PL, the sample sits directly on graphene which quenches much of the signal, so measuring the PL in situ may be quite difficult unless a sample with few layers hBN is heterostructured between WS₂ and graphene, but we do not have access to such samples currently that are of high enough quality for STM measurements. Lastly, as the excitation is expected to be in the telecom, our STM system is not set up to measure excitations within this energy range. As such, we were unable to do these measurements. Motivated by this work, we are working on incorporating Co during the growth process to create WS₂ with a low density and homogeneous distribution of Co defects in WS₂. Encapsulation between two hBN layers can then enable the measurement of both PL and coherence time. The key point we wish to convey is the ability to use HT screening to identify interesting candidates as quantum emitters. We chose atomic manipulation in combination with scanning tunneling spectroscopy, as it enables us to directly benchmark the theory, which cannot be done with optical spectroscopy (as it is only a indirect measurement).

Additionally, we are confident on the quality of the prediction we made on the optical properties of these defects especially because it is grounded on a good agreement between theory and experiment for the electronic structure. The community has high confidence in the level of theory we used as demonstrated by a series of recent predictions of spin and optical properties for defects in 2D materials using similar level of theory (e.g., the works Li et al., Tsai et al. and Lee et al.)²⁵⁻²⁷. We have clarified our statement in the line below.

We fabricate the Co_S⁰ defect, which we anticipate to exhibit brightness, a spin-doublet ground state, and a computed ZPL in the telecom at 0.966 eV, through metal deposition and subsequent sulfur vacancy substitution by cobalt with an STM tip.

Reviewer #3 (Remarks to the Author):

The Authors use High-throughput screening approach to find bright optically active defects in WS₂. They find neutral CoS⁰ (cobalt substitution to sulfur in WS₂) with a spin-doublet ground, as potentially interesting defect for optical applications, as this defect exhibit localized states within the band gap and has large transition dipole moment. The authors then use scanning tunnelling microscopy (STM) to

fabricate this defect and establish the identity of generated defect through comparison of STM results with first principles calculations. The work can of interest to the readers of Nature comm, after following revisions.

1. CoS 0 has a doublet spin ground state, so it's very unlikely (though possible) that it can be used for applications that exploit ground state spin-manipulation e.g. ODMR related applications. My question would be why Doublet defect such as this is any more favourable for quantum applications, then the singlet defects that authors found in their computational work, but chose not to fabricate those singlet defects?

Thank you for your comment. We respectfully disagree with the reviewer. Successful ODMR measurements have been demonstrated and is commonly used in in several well-known quantum defects considered as spin-photon interfaces and with doublet spin states. This includes the negatively charged SiV²⁸, the negatively charged SnV in diamond²⁹, the T center in silicon³⁰, and Vanadium in SiC³¹, etc. Those defects have doublet ground states and are actively studied as spin-photon interfaces. They have shown spin initialization and readouts and are integrated in photonic devices

2. The authors are proposing CoS 0 for optical applications probably as for single photon emission in the telecom range etc., but they have not done any specific optical characterisation, e.g. photoluminescent spectra for this defect.

Thank you for your question. We certainly want to perform optical characterization and measure the coherence time. However, this endeavor remains complex in the presented system for two reasons, 1: the way we have functionalized a defect with cobalt is one defect at a time, which we are not able to easily read out optically (we would have to find the exact same position to do a measurement on a single Co defect, and finding the exact spot where we performed STM manipulation is inherently difficult). 2: if you pull these samples out to do any optical measurements, it needs to be pulled into ambient condition and could oxidize. For these two reasons, it is difficult. Additionally, doing these types of measurements in STM with tip-assisted PL, the sample sits directly on graphene which quenches much of the signal, so measuring the PL in situ may be quite difficult unless a sample with few layers hBN is heterostructured between WS₂ and graphene, but we do not have access to such samples currently that are of high enough quality for STM measurements. Lastly, as the excitation is expected to be in the telecom, our STM system is not set up to measure excitations within this energy range. As such, we were unable do these measurements. We are now working on incorporating Co during the growth process to create WS₂ with a low density and homogeneous distribution of Co defects in WS₂. Encapsulation between two hBN layers can then enable the measurement of both PL and coherence time. This is planned future work. Additionally, we are confident on the quality of the prediction we made on the optical properties of these defects especially because it is grounded on a good agreement between theory and experiment for the electronic structure. The community has high confidence in the level of theory we used as demonstrated by a series of recent predictions of spin and optical properties for defects in 2D materials using similar level of theory (e.g., the works Li et al., Tsai et al. and Lee et al.)²⁵⁻²⁷.

3. The authors have calculated the ZPL of the CoS 0, but they did not mention how broad would the optical emission from this defect look like. What are the ΔQ , Debye-waller and Huang-Rhys factors for this optical transition?

We thank the reviewer for raising this point. We have computed the ΔQ , Debye-Waller and Huang-Rhys factors of the C_{3v} symmetry of CoS₀ using the PBE0 excited states and PBE phonons. The first excitation with ZPL of 0.96 eV results in a ΔQ of 2.47. This is larger than NV center ($\Delta Q=0.7$). This results in a large Huang-Rhys factor of 8.66, and a weak ZPL intensity because of the small Debye-Waller factor of 0.017 %. Similar values have been reported by Li and coauthors on C_S in WS₂ which has been suggested as a quantum emitter(0.003% for the first excitation) using similar level of theory²⁵.

Interestingly, we found that the second excitation of Co_S with a ZPL of 1.18 eV and a higher transition dipole moment compared to the first excitation (3.0 D), shows a much higher Debye-Waller factor of around 30%. We propose that a photonic cavity may be required to significantly enhance the zero-phonon emission^{32,33}. We hope our synthesis efforts on Co incorporated WS₂ will enable exfoliation techniques (which can be difficult to obtain clean enough substrates for highly sensitive measurements such as STM) of the doped sample that can be heterostructured between hBN for both time-resolved and conventional photoluminescence measurements. We have added the following results to the main text.

The first excitation with ZPL of 0.96 eV results in a ΔQ of 2.47, Huang-Rhys factor of 8.66, and overall results in a Debye-Waller factor of 0.017 %. Similar values have been reported by Li and coauthors on C_S in WS₂ (0.003%)²⁵. On the other hand, the second excitation of Co_S with a ZPL of 1.18 eV and a transition dipole moment 3.0 D exhibits a Debye-Waller factor of around 30%. A photonic cavity may be required to significantly enhance the zero-phonon emission^{32,33}.

I would suppose from the difference in the values of single particle KS levels and ZPL, that this emission is very broad and not narrow. If that is the case how would authors propose this defect for single photon emitting related applications?

Thank you for pointing this out. We agree with the reviewer that the difference between the single-particle KS levels and ZPL could indicate strong relaxation. However, the relaxation could originate from both electronic (e.g. vertical excitation vs. KS level differences) as well as ionic relaxation. We note that the NV center in diamond shows a large energy difference ~1 eV between the KS levels and ZPL (at the HSE level) but the NV center is the most prominent defect for quantum information science applications.

Minor comments:

1. The authors should be more specific when referring to defects with non-zero spin ground states, e.g. both doublets and triplets have “non-singlet spin multiplicity”, so authors should use terms doublets or triplets.

Thank you for pointing this out. This has been corrected where appropriate throughout the main text. Additionally, the supplementary information indicates all thermodynamically stable candidates with associated spin multiplicity.

Defect Database:

Defect	Total spin	Δ KS (eV)	TDM (Debye)
Br_{W}^0	1/2	0.854	5.79
Sc_{S}^0	1/2	0.8	3.01
Sb_{W}^-	0	1.037	6.33
Rb_{W}^-	1	0.825	10.01
Te_{W}^-	1/2	0.778	8.03
S_{W}^0	0	0.939	10.77
P_{W}^-	0	1.134	6.13
Ir_{W}^+	0	0.84	5.17
As_{W}^-	0	0.941	7.74
$\text{Pb}_{\text{W}}^{-2}$	0	1.035	6.95
C_{W}^{-2}	0	1.093	8.29
K_{W}^-	1	0.877	9.67
Ca_{W}^0	1	0.755	10.42
$\text{Ca}_{\text{W}}^{-2}$	0	0.794	9.89
Mg_{S}^+	1/2	0.786	4.23
N_{W}^-	0	1.08	9.56
Ru_{W}^0	0	0.937	3.97
Ru_{W}^+	1/2	0.824	3.06
Co_{S}^0	1/2	1.29	6.41
Bi_{W}^-	0	0.838	8.8
W_{S}^+	1/2	0.968	3.9
$\text{Vac}_{\text{W}}^{-2}$	1	0.76	7.97
Rh_{W}^-	0	0.788	7.45
Os_{W}^0	0	1.04	3.42
Fe_{S}^0	1	1.184	4.93
Sr_{W}^0	1	0.754	11.04
$\text{Sr}_{\text{W}}^{-2}$	0	0.78	10.68
Na_{W}^-	1	0.802	9.02
Zn_{S}^0	1	1.11	3.6
Ge_{S}^-	1/2	0.838	7.02
Ti_{S}^0	0	1.843	4.33
Li_{S}^0	1/2	0.764	3.8

Supplementary Table 1: All the thermodynamically stable two-level defect candidates that show transition dipole moment (TDM) larger than 2.5 D and Kohn-Sham energy difference (Δ KS) larger than 750 meV are summarized in the table below. The Δ KS is computed at single-shot PBE0 level using an α of 0.07, as detailed in the main text.

2. On a similar note, it would help if authors specifically mention how many triplets did they find?

Among 260 thermodynamically stable charged defects, we found 89 defects in singlet ground states, 94 in doublet ground states, 48 in triplet ground states, and 16 in higher states. We have now updated text to reflect on this:

... we first identified 260 charged defects that are thermodynamically stable, meaning their charge states are accessible in a certain E_F range. Of these, 89 defects exhibit singlet ground states, 94 show doublet character, 48 are triplets, and 16 are in higher states.

REFERENCES

- [1] Ye, M., Seo, H., & Galli, G. Spin coherence in two-dimensional materials. *npj Comput. Mater.* **5**, 44 (2019).
- [2] Kanai, S. et al., Generalized scaling of spin qubit coherence in over 12,000 host materials. *Proc. Natl. Acad. Sci. U.S.A.* **119**, e2121808119 (2022).
- [3] Lin, Z. et al. Defect engineering of two-dimensional transition metal dichalcogenides. *2D Mater.* **3**, 022002 (2016).
- [4] Lei, Y. et al. Graphene and beyond: Recent advances in two-dimensional materials synthesis, properties, and devices. *ACS Nanosci. Au* **2**, 450 (2022).
- [5] Kandel S. A. & Weiss, P. S. Binding and mobility of atomically resolved cobalt clusters on molybdenum disulfide. *J. Phys. Chem. B* **105**, 8102 (2001).
- [6] Tang, W. et al. Identically sized Co quantum dots on monolayer WS₂ Featuring Ohmic Contact. *Phys. Rev. Applied* **13**, 024003 (2020).
- [7] Majd, Z. G., Taghizadeh, S. F., Amiri, P., & Vaseghi, B. Half-metallic properties of transition metals adsorbed on WS₂ monolayer: A first-principles study. *J. Magn. Magn. Mater.* **481**, 129 (2019).
- [8] Xu, W. Electronic structures and magnetic properties of co-adsorbed monolayer WS₂. *J. Mater. Sci. Chem. Eng.* **4**, 32 (2016).
- [9] Schuler, B. et al. Large spin-orbit splitting of deep in-gap defect states of engineered sulfur vacancies in monolayer WS₂. *Phys. Rev. Lett.* **123**, 076801 (2019).
- [10] Subramanian, S. et al. Tuning transport across MoS₂/graphene interfaces via as-grown lateral heterostructures. *npj 2D Mater. Appl.* **4**, 9 (2020).
- [11] Feenstra, R. A prospective: Quantitative scanning tunneling spectroscopy of semiconductor surfaces. *Surf. Sci.* **603**, 2841 (2009).
- [12] Forti, S. et al. Electronic properties of single-layer tungsten disulfide on epitaxial graphene on silicon carbide. *Nanoscale* **9**, 16412 (2017).
- [13] Peng, J. et al. Human- and machine-centred designs of molecules and materials for sustainability and decarbonization. *Nat. Rev. Mater.* **7**, 991 (2022).
- [14] Freysoldt, C., Grabowski, B., Hickel, T., Neugebauer, J., Kresse, G., Janotti, A., & Van de Walle, C. G. First-principles calculations for point defects in solids. *Rev. Mod. Phys.* **86**, 253–305 (2014).
- [15] Lyons, J. L., & Van de Walle, C. G. Computationally predicted energies and properties of defects in GaN. *npj Computational Materials* **3**, 12 (2017).
- [16] Alkauskas, A., McCluskey, M. D., & Van de Walle, C. G. Tutorial: Defects in semiconductors—Combining experiment and theory. *Journal of Applied Physics* **119**, 181101 (2016).
- [17] Weber, J. R. et al. Quantum computing with defects. *Proc. Natl. Acad. Sci. U.S.A.* **107**, 8513 (2010).
- [18] Khalid, S., Medasani, B., Lyons, J. L., Wickramaratne, D., & Janotti, A. The deep-acceptor nature of the chalcogen vacancies in 2D transition-metal dichalcogenides. *2D Materials* **11**, 021001 (2024).
- [19] Bertoldo, F., Ali, S., Manti, S., & Thygesen, K. S. Quantum point defects in 2D materials - the QPOD database. *npj Computational Materials* **8**, 56 (2022).

- [20] Anderson, C. P., & Awschalom, D. D. Embracing imperfection for quantum technologies. *Phys. Today* **76**, 26 (2023).
- [21] Wolfowicz, G. et al. Quantum guidelines for solid-state spin defects. *Nat. Rev. Mater.* **6**, 906 (2021).
- [22] Hensen, B. et al. Loophole-free bell inequality violation using electron spins separated by 1.3 kilometres. *Nature* **526**, 682 (2015).
- [23] Higginbottom, D. B. et al. Optical observation of single spins in silicon. *Nature* **607**, 266 (2022).
- [24] Tang, S. et al. Quantum spin hall state in monolayer 1T'-WTe₂. *Nat. Phys.* **13**, 683 (2017).
- [25] Li, S., Thiering, G., Udvarhelyi, P., Ivády, V., & Gali, A. Carbon defect qubit in two-dimensional WS₂. *Nat. Commun.* **13**, 1 (2022).
- [26] Tsai, J.-Y., Pan, J., Lin, H., Bansil, A., & Yan, Q. Antisite defect qubits in monolayer transition metal dichalcogenides. *Nature Communications* **13**, 492 (2022).
- [27] Lee, Y., Hu, Y., Lang, X., Kim, D., Li, K., Ping, Y., Fu, K.-M. C., & Cho, K. Spin-defect qubits in two-dimensional transition metal dichalcogenides operating at telecom wavelengths. *Nature Communications* **13**, 7501 (2022).
- [28] Pingault, B. et al. Coherent control of the silicon-vacancy spin in diamond. *Nat. Commun.* **8**, 15579 (2017).
- [29] Rosenthal, E. I. et al. Microwave spin control of a tin-vacancy qubit in diamond. *Phys. Rev. X* **13**, 031022 (2023).
- [30] Higginbottom, D. B. et al. Memory and transduction prospects for silicon *T* center devices. *PRX Quantum* **4**, 020308 (2023).
- [31] Wolfowicz, G. et al. Vanadium spin qubits as telecom quantum emitters in silicon carbide. *Sci. Adv.* **6**, eaaz1192 (2020).
- [32] Schuler, B. et al. Electrically driven photon emission from individual atomic defects in monolayer WS₂. *Sci. Adv.* **6**, eabb5988 (2020).
- [33] Zhou, J. et al. Near-field coupling with a nanoimprinted probe for dark exciton nanoimaging in monolayer WSe₂. *Nano Lett.* **23**, 4901 (2023).

REVIEWERS' COMMENTS

Reviewer #1 (Remarks to the Author):

The authors have responded comprehensively to the referees' comments and I believe the manuscript is ready for publication.

Minor item: Supplementary Figure 17 caption still seems incorrect ... where is the STS from graphene shown? And panel d) is never called out.

Reviewer #2 (Remarks to the Author):

I appreciate the authors' effort in addressing all the questions. I recommend the publication of the manuscript in its current form

Reviewer #3 (Remarks to the Author):

The authors have addressed my concerns and I am happy for this paper to be published in Nat. Comm. in its current form.

REVIEWERS' COMMENTS

Reviewer #1 (Remarks to the Author):

The authors have responded comprehensively to the referees' comments and I believe the manuscript is ready for publication.

Wonderful and thank you very much for all of your help during the review process.

Minor item: Supplementary Figure 17 caption still seems incorrect ... where is the STS from graphene shown? And panel d) is never called out.

We have corrected the text in Supplementary Fig. 17, and appreciate your thorough review of our manuscript. **d** is in reference to the as-measured graphene/SiC substrate depicted in **a** (gray circle).

Supplementary Fig. 17: **Monolayer WS₂ Identification.** **a** Scanning tunneling image over a WS₂ monolayer edge resting on a graphene/SiC(0001) substrate ($I_{tunnel} = 30$ pA, $V_{sample} = 1.2$ V). Scale bar, 10 nm. **b** A height profile taken across the blue line depicted in **a**, where a height difference of ~ 0.5 nm is measured. Regions are further verified with scanning tunneling spectroscopy over both **c** as-grown WS₂ (red circle in **a**) and **d** the graphene/SiC substrate (gray circle in **a**) ($V_{modulation} = 5$ mV, $I_{set} = 150$ pA). A band gap of 2.5 eV is measured for as-grown WS₂, and graphene exhibits expected canonical band structure.

Reviewer #2 (Remarks to the Author):

I appreciate the authors' effort in addressing all the questions. I recommend the publication of the manuscript in its current form

The authors appreciate and thank the reviewer for their efforts during the review process.

Reviewer #3 (Remarks to the Author):

The authors have addressed my concerns and I am happy for this paper to be published in Nat. Comm. in its current form.

Thank you for your time and effort during the review process.